# Automated Discovery of Adaptive Attacks on Adversarial Defenses

**Chengyuan Yao**
Department of Computer Science
ETH Zürich, Switzerland
`chengyuan.yao@inf.ethz.ch`

**Pavol Bielik**
LatticeFlow
Switzerland
`pavol@latticeflow.ai`

**Petar Tsankov**
LatticeFlow
Switzerland
`petar@latticeflow.ai`

**Martin Vechev**
Department of Computer Science
ETH Zürich, Switzerland
`martin.vechev@inf.ethz.ch`

## Abstract

Reliable evaluation of adversarial defenses is a challenging task, currently limited to an expert who manually crafts attacks that exploit the defenses inner workings or approaches based on an ensemble of fixed attacks, none of which may be effective for the specific defense at hand. Our key observation is that adaptive attacks are composed of reusable building blocks that can be formalized in a search space and used to automatically discover attacks for unknown defenses. We evaluated our approach on 24 adversarial defenses and show that it outperforms `AutoAttack` (Croce & Hein, 2020b), the current state-of-the-art tool for reliable evaluation of adversarial defenses: our tool discovered significantly stronger attacks by producing 3.0%-50.8% additional adversarial examples for 10 models, while obtaining attacks with slightly stronger or similar strength for the remaining models.

## 1 Introduction

The issue of adversarial attacks (Szegedy et al., 2014; Goodfellow et al., 2015), i.e., crafting small input perturbations that lead to mispredictions, is an important problem with a large body of recent work. Unfortunately, reliable evaluation of proposed defenses is an elusive and challenging task: many defenses seem to initially be effective, only to be circumvented later by new attacks designed specifically with that defense in mind (Carlini & Wagner, 2017; Athalye et al., 2018; Tramer et al., 2020).

To address this challenge, two recent works approach the problem from different perspectives. Tramer et al. (2020) outlines an approach for manually crafting adaptive attacks that exploit the weak points of each defense. Here, a domain expert starts with an existing attack, such as `PGD` (Madry et al., 2018) (denoted as ● in Figure 1), and adapts it based on knowledge of the defense's inner workings. Common modifications include: *(i)* tuning attack parameters (e.g., number of steps), *(ii)* replacing network components to simplify the attack (e.g., removing randomization or non-differentiable components), and *(iii)* replacing the loss function optimized by the attack. This approach was demonstrated to be effective in breaking all of the considered 13 defenses. However, a downside is that it requires substantial manual effort and is limited by the domain knowledge of the expert – for instance, each of the 13 defenses came with an adaptive attack which was insufficient, in retrospect.

At the same time, Croce & Hein (2020b) proposed to assess adversarial robustness using an ensemble of four attacks illustrated in Figure 1 *(b)* – $\text{APGD}_{\text{CE}}$ with cross-entropy loss (Croce & Hein, 2020b), $\text{APGD}_{\text{DLR}}$ with difference in logit ratio loss, `FAB` (Croce & Hein, 2020a), and `SQR` (Andriushchenko et al., 2020). While these do not require manual effort and have been shown to improve the robustness

35th Conference on Neural Information Processing Systems (NeurIPS 2021).

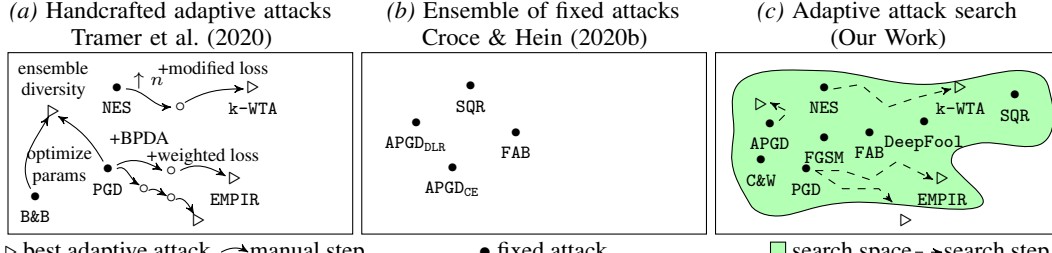

Figure 1: High-level illustration and comparison of recent works and ours. *Adaptive attacks (a)* rely on a human expert to manually adapt an existing attack to exploit the weak points of each defense. `AutoAttack (b)` evaluates defenses using an ensemble of diverse attacks. Our work *(c)* defines a search space of adaptive attacks (denoted as ▢) and performs search steps automatically.

estimate for many defenses, the approach is inherently limited by the fact that the attacks are fixed apriori without any knowledge of the given defense at hand. This is visualized in Figure 1 *(b)* where even though the attacks are designed to be diverse, they cover only a small part of the entire space.

**This work: towards automated discovery of adaptive attacks**    We present a new method that helps automating the process of crafting adaptive attacks, combining the best of both prior approaches; the ability to evaluate defenses automatically while producing attacks tuned for the given defense. Our work is based on the key observation that we can identify common techniques used to build existing adaptive attacks and extract them as reusable building blocks in a common framework. Then, given a new model with an unseen defense, we can discover an effective attack by searching over suitable combinations of these building blocks. To identify reusable techniques, we analyze existing adaptive attacks and organize their components into three groups:

- *Attack algorithm and parameters*: a library of diverse attack techniques (e.g., `APGD`, `FAB`, `C&W` (Carlini & Wagner, 2017), `NES` (Wierstra et al., 2008)), together with backbone specific and generic parameters (e.g., input randomization, number of steps, if/how to use `EOT` (Athalye et al., 2018)).

- *Network transformations*: producing an easier to attack surrogate model using techniques including variants of `BPDA` (Athalye et al., 2018) to break gradient obfuscation, and layer removal (Tramer et al., 2020) to eliminate obfuscation layers such as redundant softmax operator.

- *Loss functions*: that specify different ways of defining the attack's loss function.

These components collectively formalize an attack search space induced by their different combinations. We also present an algorithm that effectively navigates the search space so to discover an attack. In this way, domain experts are left with the creative task of designing new attacks and growing the framework by adding missing attack components, while the tool is responsible for automating many of the tedious and time-consuming trial-and-error steps that domain experts perform manually today. That is, we can automate some part of the process of finding adaptive attacks, but not necessarily the full process. This is natural as finding truly new attacks is a highly creative process that is currently out of reach for fully automated techniques.

We implemented our approach in a tool called `Adaptive AutoAttack` ($A^3$) and evaluated it on 24 diverse adversarial defenses. Our results demonstrate that $A^3$ discovers adaptive attacks that outperform `AutoAttack` (Croce & Hein, 2020b), the current state-of-the-art tool for reliable evaluation of adversarial defenses: $A^3$ finds attacks that are significantly stronger, producing 3.0%-50.8% additional adversarial examples for 10 models, while obtaining attacks with stronger or simialr strength for the remaining models. Our tool $A^3$ and scripts for reproducing the experiments are available online at:

https://github.com/eth-sri/adaptive-auto-attack

## 2 Automated Discovery of Adaptive Attacks

We use $\mathcal{D} = \{(x_i, y_i)\}_{i=1}^N$ to denote a training dataset where $x \in \mathbb{X}$ is a natural input (e.g., an image) and $y$ is the corresponding label. An adversarial example is a perturbed input $x'$, such that: *(i)* it

satisfies an attack criterion $c$, e.g., a $K$-class classification model $f \colon \mathbb{X} \to \mathbb{R}^K$ predicts a wrong label, and *(ii)* the distance $d(x', x)$ between the adversarial input $x'$ and the natural input $x$ is below a threshold $\epsilon$ under a distance metric $d$ (e.g., an $L_p$ norm). Formally, this can be written as:

$$
\text{\scshape{Adversarial}} \atop \text{\scshape{Attack}} \qquad \overbrace{d(x', x) \leq \epsilon}^{\text{capability}} \quad \text{such that} \quad \overbrace{c(f, x', x)}^{\text{goal (criterion)}}
$$

For example, instantiating this with the $L_\infty$ norm and misclassification criterion corresponds to:

$$
\text{\scshape{Misclassification}} \atop L_\infty \ \text{\scshape{Attack}} \qquad \|x' - x\|_\infty \leq \epsilon \quad \text{s.t.} \quad \hat{f}(x') \neq \hat{f}(x)
$$

where $\hat{f}$ returns the prediction $\arg\max_{k=1:K} f_k(\cdot)$ of the model $f$.

**Problem Statement**  Given a model $f$ equipped with an unknown set of defenses and a dataset $\mathcal{D} = \{(x_i, y_i)\}_{i=1}^N$, our goal is to find an adaptive adversarial attack $a \in \mathcal{A}$ that is best at generating adversarial samples $x'$ according to the attack criterion $c$ and the attack capability $d(x', x) \leq \epsilon$:

$$
\max_{a \in \mathcal{A}, \ d(x', x) \leq \epsilon} \mathbb{E}_{(x,y) \sim \mathcal{D}} \quad c(f, x', x) \qquad \text{where} \quad x' = a(x, f) \tag{1}
$$

Here, $\mathcal{A}$ denotes the search space of all possible attacks, where the goal of each attack $a \colon \mathbb{X} \times (\mathbb{X} \to \mathbb{R}^K) \to \mathbb{X}$ is to generate an adversarial sample $x' = a(x, f)$ for a given input $x$ and model $f$. For example, solving this optimization problem with respect to the $L_\infty$ misclassification criterion corresponds to optimizing the number of adversarial examples misclassified by the model.

In our work, we consider an *implementation-knowledge adversary*, who has full access to the model's implementation at inference time (e.g., the model's computational graph). We chose this threat model as it matches our problem setting – given an unseen model implementation, we want to automatically find an adaptive attack that exploits its weak points but without the need of a domain expert. We note that this threat model is weaker than a *perfect-knowledge adversary* (Biggio et al., 2013), which assumes a domain expert that also has knowledge about the training dataset and algorithm, as this information is difficult, or even not possible, to recover from the model's implementation only.

**Key Challenges**  To solve the optimization problem from Eq. 1, we address two key challenges:

- *Defining a suitable attacks search space $\mathcal{A}$* such that it is expressible enough to cover a range of existing adaptive attacks.

- *Searching over the space $\mathcal{A}$ efficiently* such that a strong attack is found within a reasonable time.

Next, we formalize the attack space in Section 3 and then describe our search algorithm in Section 4.

## 3  Adaptive Attacks Search Space

We define the adaptive attack search space by analyzing existing adaptive attacks and identifying common techniques used to break adversarial defenses. Formally, the adaptive attack search space is given by $\mathcal{A} \colon \mathbb{S} \times \mathbb{T}$, where $\mathbb{S}$ consists of sequences of backbone attacks along with their loss functions, selected from a space of loss functions $\mathbb{L}$, and $\mathbb{T}$ consists of network transformations. Semantically, given an input $x$ and a model $f$, the goal of adaptive attack $(s, t) \in \mathbb{S} \times \mathbb{T}$ is to return an adversarial example $x'$ by computing $s(x, t(f)) = x'$. That is, it first transforms the model $f$ by applying the transformation $t(f) = f'$, and then executes the attack $s$ on the surrogate model $f'$. Note that the surrogate model is used only to compute the candidate adversarial example, not to evaluate it. That is, we generate an adversarial example $x'$ for $f'$, and then check whether it is also adversarial for $f$. Since $x'$ may be adversarial for $f'$, but not for $f$, the adaptive attack must maximize the transferability of the generated candidate adversarial samples.

**Attack Algorithm & Parameters ($\mathbb{S}$)**  The attack search space consists of a sequence of adversarial attacks. We formalize the search space with the grammar:

```
(Attack Search Space)
    𝕊 ::=  𝕊; 𝕊 | randomize 𝕊 | EOT 𝕊, n | repeat 𝕊, n |
           try 𝕊 for n | Attack with params with loss ∈ 𝕃
```

- $\mathbb{S}$; $\mathbb{S}$: composes two attacks, which are executed independently and return the first adversarial sample in the defined order. That is, given input $x$, the attack $s_1; s_2$ returns $s_1(x)$ if $s_1(x)$ is an adversarial example, and otherwise it returns $s_2(x)$.

- randomize $\mathbb{S}$: enables the attack's randomized components, which correspond to random seed and/or selecting a starting point within $d(x', x) \leq \epsilon$, uniformly at random.

- EOT $\mathbb{S}$, n: uses expectation over transformation, a technique designed to compute gradients for models with randomized components (Athalye et al., 2018).

- repeat $\mathbb{S}$, n: repeats the attack $n$ times (useful only if randomization is enabled).

- try $\mathbb{S}$ for n: executes the attack with a time budget of n seconds.

- Attack with params with loss ∈ $\mathbb{L}$: is a backbone attack Attack executed with parameters params and loss function loss. In our evaluation, we use FGSM (Goodfellow et al., 2015), PGD, DeepFool (Moosavi-Dezfooli et al., 2016), C&W, NES, APGD, FAB and SQR. We provide full list of the attack parameters, including their ranges and priors in Appendix B.

Note, that we include variety of backbone attacks, including those that were already superseeded by stronger attacks. This is done for two key reasons. First, weaker attacks can be surprisingly effective in some cases and avoid the detector because of their weakness (see defense C24 in our evaluation). Second, we are not using any prior when designing the space search. In particular, whenever a new attack is designed it can simply be added to the search space. Then, the goal of the search algorithm is to be powerful enough to perform the search efficiently. In other words, the aim is to avoid making any assumptions of what is useful or not and let the search algorithm learn this instead.

**Network Transformations ($\mathbb{T}$)**   A common approach that aims to improve the robustness of neural networks against adversarial attacks is to incorporate an explicit defense mechanism in the neural architecture. These defenses often obfuscate gradients to render iterative-optimization methods ineffective (Athalye et al., 2018). However, these defenses can be successfully circumvented by *(i)* choosing a suitable attack algorithm, such as score and decision-based attacks (included in $\mathbb{S}$), or *(ii)* by changing the neural architecture (defined next).

At a high level, the network transformation search space $\mathbb{T}$ takes as input a model $f$ and transforms it to another model $f'$, which is easier to attack. To achieve this, the network $f$ can be expressed as a directed acyclic graph, including both the forward and backward computations, where each vertex denotes an operator (e.g., convolution, residual blocks, etc.), and edges correspond to data dependencies. In our work, we include two types of network transformations:

*Layer Removal*, which removes an operator from the graph. Each operator can be removed if its input and output dimensions are the same, regardless of its functionality.

*Backward Pass Differentiable Approximation* (BPDA) (Athalye et al., 2018), which replaces the backward version of an operator with a differentiable approximation of the function. In our search space we include three different function approximations: *(i)* an identity function, *(ii)* a convolution layer with kernel size 1, and *(iii)* a two-layer convolutional layer with ReLU activation in between. The weights in the latter two cases are learned through approximating the forward function.

**Loss Function ($\mathbb{L}$)**   Selecting the right objective function to optimize is an important design decision for creating strong adaptive attacks. Indeed, the recent work of Tramer et al. (2020) uses 9 different objective functions to break 13 defenses, showing the importance of this step. We formalize the space of possible loss functions using the following grammar:

```
(Loss Function Search Space)
   𝕃 ::=  targeted Loss, n with Z | untargeted Loss with Z |
          targeted Loss, n − untargeted Loss with Z
   Z ::=  logits | probs
Loss ::=  CrossEntropy | HingeLoss | L1 | DLR | LogitMatching
```

*Targeted vs Untargeted.* The loss can be either untargeted, where the goal is to change the classification to any other label $f(x') \neq f(x)$, or targeted, where the goal is to predict a concrete label $f(x') = l$. Even though the untargeted loss is less restrictive, it is not always easier to optimize in practice, and replacing it with a targeted attack might perform better. When using `targeted` `Loss, n`, the attack will consider the top `n` classes with the highest probability as the targets.

*Loss Formulation.* The concrete loss formulation includes loss functions used in existing adaptive attacks, as well as the recently proposed difference in logit ratio loss (Croce & Hein, 2020b). We provide a formal definition of the loss functions used in our work in Appendix B.

*Logits vs. Probabilities.* In our search space, loss functions can be instantiated both with logits as well as with probabilities. Note that some loss functions are specifically designed for one of the two options, such as `C&W` (Carlini & Wagner, 2017) or `DLR` (Croce & Hein, 2020b). While such knowledge can be used to reduce the search space, it is not necessary as long as the search algorithm is powerful enough to recognize that such a combination leads to poor results.

*Loss Replacement.* Because the key idea behind many of the defenses is finding a property that helps differentiate between adversarial and natural images, one can also define the optimization objective in the same way. These feature-level attacks (Sabour et al., 2016) avoid the need to directly optimize the complex objective defined by the adversarial defense and have been effective at circumventing them. As an example, the logit matching loss minimizes the difference of logits between adversarial sample $x'$ and a natural sample of the target class $x$ (selected at random from the dataset). Instead of logits, the same idea can also be applied to other statistics, such as internal representations computed by a pre-trained model or KL-divergence between label probabilities.

# 4   Search Algorithm

We now describe our search algorithm that optimizes the problem statement from Eq. 1. Since we do not have access to the underlying distribution, we approximate Eq. 1 using the dataset $\mathcal{D}$ as follows:

$$score(f, a, \mathcal{D}) = \frac{1}{|\mathcal{D}|} \sum_{i=1}^{|\mathcal{D}|} -\lambda l_a + \max_{d(x', x) \leq \epsilon} c(f, a(x, f), x) \tag{2}$$

where $a \in \mathcal{A}$ is an attack, $l_a \in \mathbb{R}^+$ denotes untargeted cross-entropy loss of $a$ on the input, and $\lambda \in \mathbb{R}$ is a hyperparameter. The intuition behind $-\lambda \cdot l_a$ is that it acts as a tie-breaker in case the criterion $c$ alone is not enough to differentiate between multiple attacks. While this is unlikely to happen when evaluating on large datasets, it is quite common when using only a small number of samples. Obtaining good estimates in such cases is especially important for achieving scalability since performing the search directly on the full dataset would be prohibitively slow.

**Search Algorithm**   We present our search algorithm in Algorithm 1. We start by searching through the space of network transformations $t \in \mathbb{T}$ to find a suitable surrogate model (line 1). This is achieved by taking the default attack $\Delta$ (in our implementation, we set $\Delta$ to $\text{APGD}_{\text{CE}}$), and then evaluating all locations where BPDA can be used, and subsequently evaluating all layers that can be removed. Even though this step is exhaustive, it takes only a fraction of the runtime in our experiments, and no further optimization was necessary.

Next, we search through the space of attacks $\mathbb{S}$. As this search space is enormous, we employ three techniques to improve scalability and attack quality. First, to generate a sequence of $m$ attacks, we perform a greedy search (lines 3-16). That is, in each step, we find an attack with the best score on the samples not circumvented by any of the previous attacks (line 4). Second, we use a parameter estimator model $M$ to select the suitable parameters (line 7). In our work, we use Tree of Parzen Estimators (Bergstra et al., 2011), but the concrete implementation can vary. Once the parameters are selected, they are evaluated using the *score* function (line 8), the result is stored in the trial history $\mathcal{H}$ (line 9), and the estimator is updated (line 10). Third, because evaluating the adversarial attacks can be expensive, and the dataset $\mathcal{D}$ is typically large, we employ successive halving technique (Karnin et al., 2013; Jamieson & Talwalkar, 2016). Concretely, instead of evaluating all the trials on the full dataset, we start by evaluating them only on a subset of samples $\mathcal{D}_{\text{sample}}$ (line 5). Then, we improve the score estimates by iteratively increasing the dataset size (line 13), re-evaluating the scores (line 14), and retaining a quarter of the trials with the best score (line 15). We repeat this process to find a single best attack from $\mathcal{H}$, which is then added to the sequence of attacks $\mathcal{S}$ (line 16).

**Algorithm 1:** A search algorithm that given a model $f$ with unknown defense, discovers an adaptive attack from the attack search space $\mathcal{A}$ with the best *score*.

**def** `AdaptiveAttackSearch`

    **Input:** dataset $\mathcal{D}$, model $f$, attack search space $\mathcal{A} = \mathbb{S} \times \mathbb{T}$, number of trials $k$, initial dataset size $n$, attack sequence length $m$, criterion function $c$, initial parameter estimator model $M$, default attack $\Delta \in \mathbb{S}$

    **Output:** adaptive attack from $a_{[s,t]} \in \mathcal{A} = \mathbb{S} \times \mathbb{T}$ achieving the highest *score* on $\mathcal{D}$

1    $t \leftarrow \arg\max_{t \in \mathbb{T}} score(f, a_{[\Delta,t]}, \mathcal{D})$      ▷ Find surrogate model $t$ using default attack $\Delta$

2    $\mathcal{S} \leftarrow \bot$                  ▷ Initialize attack to be no attack, which returns the input image

3    **for** $j \leftarrow 1:m$ **do**                ▷ Run $m$ iterations to get sequence of $m$ attacks

4      $\mathcal{D} \leftarrow \mathcal{D} \setminus \{x \mid x \in \mathcal{D} \wedge c(f, a_{[\mathcal{S},t]}(x,f), x)\}$      ▷ Remove non-robust samples

5      $\mathcal{H} \leftarrow \emptyset; \mathcal{D}_{\texttt{sample}} \leftarrow sample(\mathcal{D}, n)$      ▷ Initial dataset with $n$ samples

6      **for** $i \leftarrow 1:k$ **do**           ▷ Select candidate adaptive attacks

7          $\theta' \leftarrow \arg\max_{\theta \in \mathbb{S}} P(\theta \mid M)$ ▷ Best unseen parameters according to the model $M$

8          $q \leftarrow score(f, a_{[\theta',t]}, \mathcal{D}_{\texttt{sample}})$

9          $\mathcal{H} \leftarrow \mathcal{H} \cup \{(\theta', q)\}$

10         $M \leftarrow$ update model $M$ with $(\theta', q)$

11      $\mathcal{H} \leftarrow$ keep $|\mathcal{H}|/4$ attacks with the best score

12      **while** $|\mathcal{H}| > 1$ **and** $\mathcal{D}_{sample} \neq \mathcal{D}$ **do**      ▷ Successive halving (SHA)

13          $\mathcal{D}_{\texttt{sample}} \leftarrow \mathcal{D}_{\texttt{sample}} \cup sample(\mathcal{D} \setminus \mathcal{D}_{\texttt{sample}}, |\mathcal{D}_{\texttt{sample}}|)$ ▷ Double the dataset size

14          $\mathcal{H} \leftarrow \{(\theta, score(f, a_{[\theta,t]}, \mathcal{D}_{\texttt{sample}})) \mid (\theta, q) \in \mathcal{H}\}$ ▷ Re-evaluate on larger dataset

15          $\mathcal{H} \leftarrow$ keep $|\mathcal{H}|/4$ attacks with the best score

16      $\mathcal{S} \leftarrow \mathcal{S}$; best attack in $\mathcal{H}$

17    **return** $a_{[\mathcal{S},t]}$

**Time Budget and Worst-case Search Time** We set a time budget on the attacks, measured in seconds per sample per attack, to restrict resource-expensive attacks and allow the tradeoff between computation time and attack strength. If an attack exceeds the time limit in line 8, the evaluation terminates, and the score is set to $-\infty$. We analyzed the worst-case search time to be $4/3\times$ the allowed attack runtime in our experiments, which means the search overhead is both controllable and reasonable in practice. The derivation is shown in Appendix A.

## 5 Evaluation

We evaluate $\texttt{A}^3$ on 24 models with diverse defenses and compare the results to $\texttt{AutoAttack}$ (Croce & Hein, 2020b) and to several existing handcrafted attacks. $\texttt{AutoAttack}$ is a state-of-the-art tool designed for reliable evaluation of adversarial defenses that improved the originally reported results for many existing defenses by up to 10%. Our key result is that $\texttt{A}^3$ finds stronger or similar attacks than $\texttt{AutoAttack}$ for virtually all defenses:

- In 10 cases, the attacks found by $\texttt{A}^3$ are significantly stronger than $\texttt{AutoAttack}$, resulting in 3.0% to 50.8% additional adversarial examples.

- In the other 14 cases, $\texttt{A}^3$'s attacks are typically 2x faster while enjoying similar attack quality.

**Model Selection** We selected 24 models and divided them into three blocks A, B, C as listed in Table 1. Block A contains diverse defenses with $\epsilon = 4/255$. Block B contains selected top models from $\texttt{RobustBench}$ (Croce et al., 2020). Block C contains diverse defenses with $\epsilon = 8/255$.

**The $\texttt{A}^3$ tool** The implementation of $\texttt{A}^3$ is based on $\texttt{PyTorch}$ (Paszke et al., 2019), the implementations of $\texttt{FGSM}$, $\texttt{PGD}$, $\texttt{NES}$, and $\texttt{DeepFool}$ are based on $\texttt{FoolBox}$ (Rauber et al., 2017) version 3.0.0, $\texttt{C\&W}$ is based on $\texttt{ART}$ (Nicolae et al., 2018) version 1.3.0, and the attacks $\texttt{APGD}$, $\texttt{FAB}$, and $\texttt{SQR}$ are from (Croce & Hein, 2020b). We use $\texttt{AutoAttack}$'s *rand* version if a defense has a randomization component, and otherwise we use its *standard* version. To allow for a fair comparison, we extended $\texttt{AutoAttack}$ with backward pass differential approximation (BPDA), so we can run it on defenses with non-differentiable components; without this, all gradient-based attacks would fail. We instantiate

Table 1: Comparison of `AutoAttack` (AA) and our approach ($A^3$) on 24 defenses. Further details of each defense, discovered adaptive attacks and confidence intervals are included in Appendix D and H.

| | | Robust Accuracy (1 - Rerr) | | | Runtime (min) | | | Search |
|---|---|---|---|---|---|---|---|---|
| **CIFAR-10, $l_\infty$, $\epsilon = 4/255$** | | AA | $A^3$ | $\Delta$ | AA | $A^3$ | Speed-up | $A^3$ |
| A1 | Madry et al. (2018) | 44.78 | **44.69** | -0.09 | 25 | 20 | 1.25× | 88 |
| A2[†] | Buckman et al. (2018) | 2.29 | **1.96** | -0.33 | 9 | 7 | 1.29× | 116 |
| A3[†] | Das et al. (2017) + Lee et al. (2018) | 0.59 | **0.11** | -0.48 | 6 | 2 | 3.00× | 40 |
| A4 | Metzen et al. (2017) | 6.17 | **3.04** | -3.13 | 21 | 13 | 1.62× | 80 |
| A5 | Guo et al. (2018) | 22.30 | **12.14** | -10.16 | 19 | 17 | 1.12× | 99 |
| A6[†] | Pang et al. (2019) | 4.14 | **3.94** | -0.20 | 28 | 24 | 1.17× | 237 |
| A7 | Papernot et al. (2015) | 2.85 | **2.71** | -0.14 | 4 | 4 | 1.00× | 84 |
| A8 | Xiao et al. (2020) | 19.82 | **11.11** | -8.71 | 49 | 22 | 2.23× | 189 |
| A9 | Xiao et al. (2020)$_{ADV}$ | 64.91 | **63.56** | -1.35 | 157 | 100 | 1.57× | 179 |
| A9' | Xiao et al. (2020)$_{ADV}$ | 64.91 | **17.70** | -47.21 | 157 | 2,280 | 0.07× | 1,548 |
| **CIFAR-10, $l_\infty$, $\epsilon = 8/255$** | | | | | | | | |
| B10[*] | Gowal et al. (2021) | 62.80 | **62.79** | -0.01 | 818 | 226 | 3.62× | 761 |
| B11[*] | Wu et al. (2020)$_{RTS}$ | 60.04 | **60.01** | -0.03 | 706 | 255 | 2.77× | 690 |
| B12[*] | Zhang et al. (2021) | 59.64 | **59.56** | -0.08 | 604 | 261 | 2.31× | 565 |
| B13[*] | Carmon et al. (2019) | 59.53 | **59.51** | -0.02 | 638 | 282 | 2.26× | 575 |
| B14[*] | Sehwag et al. (2020) | **57.14** | 57.16 | 0.02 | 671 | 429 | 1.56× | 691 |
| **CIFAR-10, $l_\infty$, $\epsilon = 8/255$** | | | | | | | | |
| C15[*] | Stutz et al. (2020) | 77.64 | **39.54** | -38.10 | 101 | 108 | 0.94× | 296 |
| C15' | Stutz et al. (2020) | 77.64 | **26.87** | -50.77 | 101 | 205 | 0.49× | 659 |
| C16[*] | Zhang & Wang (2019) | **36.74** | 37.11 | 0.37 | 381 | 302 | 1.26× | 726 |
| C17 | Grathwohl et al. (2020) | **5.15** | 5.16 | 0.01 | 107 | 114 | 0.94× | 749 |
| C18 | Xiao et al. (2020)$_{ADV}$ | 5.40 | **2.31** | -3.09 | 95 | 146 | 0.65× | 828 |
| C19 | Wang et al. (2019) | 50.84 | **50.81** | -0.03 | 734 | 372 | 1.97× | 755 |
| C20[†] | B11 + Defense in A3 | 60.72 | **60.04** | -0.68 | 621 | 210 | 2.96× | 585 |
| C21[†] | C17 + Defense in A3 | 15.27 | **5.24** | -10.03 | 261 | 79 | 3.30× | 746 |
| C22 | B11 + Random Rotation | 49.53 | **41.99** | -7.54 | 255 | 462 | 0.55× | 900 |
| C23 | C17 + Random Rotation | 22.29 | **13.45** | -8.84 | 114 | 374 | 0.30× | 1,023 |
| C24 | Hu et al. (2019) | 6.25 | **3.07** | -3.18 | 110 | 56 | 1.96× | 502 |

[*]model available from the authors, [†]model with non-differentiable components.

B12 uses $\epsilon = 0.031$. C15 uses $\epsilon = 0.03$. A9' uses time budget $T_c = 30$. C15' uses $m = 8$.

Algorithm 1 by setting: the attack sequence length $m = 3$, the number of trials $k = 64$, the initial dataset size $n = 100$, and we use a time budget of $0.5$ to $3$ seconds per sample depending on the model size. All of the experiments are performed using a single RTX 2080 Ti GPU.

**Evaluation Metric** Following Stutz et al. (2020), we use the *robust test error (Rerr)* metric to combine the evaluation of defenses with and without detectors. We include details in Appendix C. In our evaluation, $A^3$ produces consistent results on the same model across independent runs with the standard deviation $\sigma < 0.2$ (computed across 3 runs). The details are included in Appendix H.

**Comparison to `AutoAttack`** Our main results, summarized in Table 1, show the robust accuracy (lower is better) and runtime of both `AutoAttack` (AA) and $A^3$ over the 24 defenses. For example, for A8 our tool finds an attack that leads to lower robust accuracy (11.1% for $A^3$ vs. 19.8% for AA) and is more than twice as fast (22 min for $A^3$ vs. 49 min for AA). Overall, $A^3$ significantly improves upon AA or provides similar but faster attacks.

We note that the attacks from AA are included in our search space (although without the knowledge of their best parameters and sequence), and so it is expected that $A^3$ performs at least as well as AA, provided sufficient exploration time. Importantly, $A^3$ often finds better attacks: for 10 defenses, $A^3$ reduces the robust accuracy by 3% to 50% compared to AA. Next, we discuss the results in more detail.

*Defenses based on Adversarial Training.* Models in block B are selected from `RobustBench` (Croce et al., 2020), and they are based on various extensions of adversarial training, such as using additional unlabelled data for training, extensive hyperparameter tuning, instance weighting or loss regularization. The results show that the robustness reported by AA is already very high and using $A^3$ leads to only marginal improvement. However, because our tool also optimizes for the runtime, $A^3$ does achieve significant speed-ups, ranging from $1.5\times$ to $3.6\times$. The reasons behind the marginal robustness improvement of $A^3$ are two-fold. First, it shows that $A^3$ is limited by the attack techniques search space, as the attack found are all variations of APGD. Second, the models B10 - B14 aim to improve the adversarial training procedure rather than developing a new defence. This is in contrast to models that do design various types of new defences (included in blocks A and C), evaluating which typically requires discovering a new adaptive attack. For these new defences, evaluation is much more difficult and this is where our approach also improves the most.

*Obfuscation Defenses.* Defenses A3, A8, A9, C18, C20, and C21 are based on gradient obfuscation. $A^3$ discovers stronger attacks that reduce the robust accuracy for all defenses by up to 47.21%. Here, removing the obfuscated defenses in A3, C20, and C21 provides better gradient estimation for the attacks. Further, the use of more suitable loss functions strengthens the discovered attacks and improves the evaluation results for A8 and C18.

*Randomized Defenses.* For the randomized input defenses A8, C22, and C23, $A^3$ discovers attacks that, compared to AA's *rand* version, further reduce robustness by 8.71%, 7.54%, and 8.84%, respectively. This is achieved by using stronger yet more costly parameter settings, attacks with different backbones (APGD, PGD) and 7 different loss functions (as listed in Appendix D).

*Detector based Defenses.* For C15, A4, and C24 defended with detectors, $A^3$ improves over AA by reducing the robustness by 50.77%, 3.13%, and 3.18%, respectively. This is because none of the attacks discovered by $A^3$ are included in AA. Namely, $A^3$ found $SQR_{DLR}$ and $APGD_{Hinge}$ for C15, untargeted FAB for A4 (FAB in AA is targeted), and $PGD_{L1}$ for C24.

**Generalization of $A^3$**   Given a new defense, the main strength of our approach is that it directly benefits from all existing techniques included in the search space. Here, we compare our approach to three handcrafted adaptive attacks not included in the search space.

As a first example, C15 (Stutz et al., 2020) proposes an adaptive attack PGD-Conf with backtracking that leads to robust accuracy of 36.9%, which can be improved to 31.6% by combining PGD-Conf with blackbox attacks. $A^3$ finds $APGD_{Hinge}$ and Z = probs. This combination is interesting since the hinge loss maximizing the difference between the top two predictions, in fact, reflects the PGD-Conf objective function. Further, similarly to the manually crafted attack by C15, a different blackbox attack included in our search space, $SQR_{DLR}$, is found to complement the strength of APGD. When using a sequence of three attacks, we achieve 39.54% robust accuracy. We can decrease the robust accuracy even further by increasing the number of attacks to eight – the robust accuracy drops to 26.87%, which is a stronger result than the one reported in the original paper. In this case, our search space and the search algorithm are powerful enough to not only replicate the main ideas of Stutz et al. (2020) but also to improve its evaluation when allowing for a larger attack budget. Note that this improvement is possible even without including the backtracking used by PGD-Conf as a building block in our search space. In comparison, the robust accuracy reported by AA is only 77.64%.

As a second example, C18 is known to be susceptible to NES which achieves 0.16% robust accuracy (Tramer et al., 2020). To assess the quality of our approach, we remove NES from our search space and instead try to discover an adaptive attack using the remaining building blocks. In this case, our search space was expressive enough to find an alternative attack that achieves 2.31% robust accuracy.

As a third example, to break C24, Tramer et al. (2020) designed an adaptive attack that linearly interpolates between the original and the adversarial samples using PGD. This technique breaks the defense and achieves 0% robust accuracy. In comparison, we find $PGD_{L1}$, which achieves 3.07% robust accuracy. In this case, the fact that $PGD_{L1}$ is a relatively weak attack is an advantage – it successfully bypasses the detector by not generating overconfident predictions.

**$A^3$ Interpretability**   As illustrated above, it is possible to manually analyze the discovered attacks in order to understand how they break the defense mechanism. Further, we can also gain insights from the patterns of attacks searched across all the models (shown in Appendix D, Table 6). For example, it turns out that $\ell_{CE}$ is not as frequent as $\ell_{DLR}$ or $\ell_{hinge}$. This fact challenges the common

practice of using $\ell_{CE}$ as the default loss when evaluating robustness. In addition, using $\ell_{CE}$ during adversarial training can make models resilient to $\ell_{CE}$, loss, but not necessarily to other losses.

$\mathtt{A}^3$ **Scalability** To assess $\mathtt{A}^3$'s scalability, we perform two ablation studies: *(i)* increase the search space by $4\times$ (by adding 8 random attacks, their corresponding parameters, and 4 dummy losses), and *(ii)* keep the search space size unchanged but reduce the search runtime by half. In *(i)*, we observed a marginal performance decrease when using the same runtime, and we can reach the same attack strength when the runtime budget is increased by $1.5\times$. In *(ii)*, even when we reduce the runtime by half, we can still find attacks that are only slightly worse ($\le 0.4$). This shows that a budget version of the search can provide a strong robustness evaluation. We include detailed results in Appendix E.

**Ablation Studies** Similar to existing handcrafted adaptive attacks, all three components included in the search space were important for generating strong adaptive attacks for a variety of defenses. Here we briefly discuss their importance while including the full experiment results in Appendix F.

*Attack & Parameters.* We demonstrate the importance of parameters by comparing `PGD`, `C&W`, `DF`, and `FGSM` with default library parameters to the best configuration found when available parameters are included in the search space. The attacks found by $\mathtt{A}^3$ are on average 5.5% stronger than the best attack among the four attacks on `A` models.

*Loss Formulation.* To evaluate the effect of modeling different loss functions, we remove them from the search space and keep only the original loss function defined for each attack. The search score drops by 3% on average for `A` models without the loss formulation.

*Network Processing.* In `C21`, the main reason for achieving 10% decrease in robust accuracy is the removal of the gradient obfuscated defense Reverse Sigmoid. We provide a more detailed ablation in Table 2, which shows the effect of different `BPDA` instantiations included in our search space. For `A2`, since the non-differentiable layer is non-linear thermometer encoding, it is better to use a function with non-linear activation to approximate it. For `A3`, `C20`, `C21`, the defense is image JPEG compression and identity network is the best algorithm since the networks can overfit when training on limited data.

Table 2: The robust accuracy (1 - Rerr) of networks with different BPDA policies evaluated by $\mathtt{APGD}_{CE}$ with 50 iterations.

| BPDA Type | A2 | A3 | C20 | C21 |
|---|---|---|---|---|
| identity | 18.5 | **9.6** | **70.5** | **84.0** |
| 1x1 convolution | 8.9 | 10.3 | 70.8 | 84.9 |
| 2 layer conv+ReLU | **3.7** | 14.9 | 74.1 | 86.2 |

## 6  Related Work

The most closely related work to ours is `AutoAttack` (Croce & Hein, 2020b), which improves the evaluation of adversarial defenses by proposing an ensemble of four fixed attacks. Further, the key to stronger attacks was a new algorithm `APGD`, which improves upon `PGD` by halving the step size dynamically based on the loss at each step. In our work, we improve over `AutoAttack` in three keys aspects: *(i)* we formalize a search space of adaptive attacks, rather than using a fixed ensemble, *(ii)* we design a search algorithm that discovers the best adaptive attacks automatically, significantly improving over the results of `AutoAttack`, and *(iii)* our search space is extensible and allows reusing building blocks from one attack by other attacks, effectively expressing new attack instantiations. For example, the idea of dynamically adapting the step size is not tied to `APGD`, but it is a general concept applicable to any step-based algorithm.

Another related work is Composite Adversarial Attacks (`CAA`) (Mao et al., 2021). The main idea of `CAA` is that instead of selecting an ensemble of four attacks that complement each other as done by `AutoAttack`, the authors proposed to search for a sequence of attacks that achieve the best performance. Here, the authors focus on evaluating defences based on adversarial training and show improvements of up to 1% over `AutoAttack`. In comparison, our main idea is that the way adaptive attacks are designed today can be formalized as a search space that includes not only sequence of attacks but also loss functions, network processing and rich space of hyperparameters. This is critical as it defines a much larger search space to cover a wide range of defenses, beyond the reach of both `CA` and `AutoAttack`. This can be also seen in our evaluation – we achieve significant improvement by finding 3% to 50% more adversarial examples for 10 models.

Our work is also closely related to the recent advances in AutoML, such as in the domain of neural architecture search (NAS) (Zoph & Le, 2017; Elsken et al., 2019). Similar to our work, the core challenge in NAS is an efficient search over a large space of parameters and configurations, and therefore many techniques can also be applied to our setting. This includes BOHB (Falkner et al., 2018), ASHA (Li et al., 2018), using gradient information coupled with reinforcement learning (Zoph & Le, 2017) or continuous search space formulation (Liu et al., 2019). Even though finding completely novel neural architectures is often beyond the reach, NAS is still very useful and finds many state-of-the-art models. This is also true in our setting – while human experts will continue to play a key role in defining new types of adaptive attacks, as we show in our work, it is already possible to automate many of the intermediate steps.

## 7   Conclusion

We presented the first tool that aims to automate the process of finding strong adaptive attacks specifically tailored to a given adversarial defense. Our key insight is that we can identify reusable techniques used in existing attacks and formalize them into a search space. Then, we can phrase the challenge of finding new attacks as an optimization problem of finding the strongest attack over this search space.

Our approach automates the tedious and time-consuming trial-and-error steps that domain experts perform manually today, allowing them to focus on the creative task of designing new attacks. By doing so, we also immediately provide a more reliable evaluation of new and existing defenses, many of which have been broken only after their proposal because the authors struggled to find an effective attack by manually exploring the vast space of techniques. Importantly, even though our current search space contains only a subset of existing techniques, our evaluation shows that $A^3$ can partially re-discover or even improve upon some handcrafted adaptive attacks not included in our search space.

However, there are also limitations to overcome in future work. First, while the search space can be easily extended, it is also inherently incomplete, and domain experts will still play an important role in designing novel types of attacks. Second, the search algorithm does not model the attack runtime and as a result, incorporating expensive attacks can be computational unaffordable. This is problematic as it can incur huge overhead even if a fast attack does exist. Finally, an interesting future work is to use meta-learning to improve the search even further, allowing $A^3$ to learn across multiple models, rather than starting each time from scratch.

## 8   Societal Impacts

In this paper, an approach to improve the evaluation of adversarial defenses by automatically finding adaptive adversarial attacks is proposed and evaluated. As such, this work builds on a large body of existing research on developing adversarial attacks and defences and thus shares the same societal impacts. Concretely, the presented approach can be used both in a beneficial way by the researchers developing adversarial defenses, as well as, in a malicious way by an attacker trying to break existing models. In both cases, the approach is designed to improve empirical model evaluation, rather than providing verified model robustness, and thus is not intended to provide formal robustness guarantees for safety-critical applications. For applications where formal robustness guarantess are required, instead of using empirical techniques as in this work, one should instead adapt the concurrent line of work on certified robustness.

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
