# A  A³ Time Complexity

This section gives the worst-case time analysis for Algorithm 1. We denote $T_a$ to be the attack time and $T_r$ to be the search time. We will show that with the per sample per attack time constraint of $T_c$:

$$T_a \leq m \times N \times T_c \tag{3}$$

$$T_r \leq 2 \times m \times n \times k \times T_c \tag{4}$$

Where $m$, $N$, $n$, $k$ are the number of attacks, the size of the dataset $D$, the size of initial dataset size, the number of attacks to sample respectively.

In Algorithm 1, only steps on lines 1,4,8,14 are timing critical as they apply the expensive attack algorithms. Other steps like sampling datasets and applying parameter estimator $M$ are considered as constant overhead. $T_a$ is the total runtime of line 4, because line 4 is the step to apply the attack on all the samples. $T_r$ includes the runtime of lines 1,8,14.

$T_a$ has the worst-case runtime when each of the $m$ attacks uses the full time budget $T_c$ on all the samples (denoted as $N$). This gives the bound shown in Eq. 3.

For $T_r$, we first analyze the time in lines 8 and 14 for a single attack. In line 8, the maximum time to perform $k$ attacks on $n$ samples is: $n \times k \times T_c$. In line 14, the cost of the first iteration is: $\frac{1}{2} n \times k \times T_c$ as there are $k/4$ attacks and $2n$ samples. The cost of SHA iteration is halved for every subsequent iteration by such design, so the total time for line 14 is $n \times k \times T_c$. As there are $m$ attacks, the total time bound for lines 8 and 14 is: $2 \times m \times n \times k \times T_c$.

The runtime for line 1 is bounded by $N \times T_{fast}$ as we run single attack on all the samples. Here, we use $T_{fast}$ to denote the maximum runtime of a fast attack that we run at this stage. This step is typically negligible compared to the subsequent search, i.e., $N \times T_{fast} \ll 2 \times m \times n \times k \times T_c$. Overall, we can therefore bound the search runtime by considering the lines and 8 and 14, which leads to the bound from Eq. 4.

In our evaluation, we use $m = 3, k = 64, n = 100, N = 10000$. Substituting into Eq. 4 leads to $T_r \leq 2 \times 3 \times 100 \times 64 \times T_c \leq 4 \times N \times T_c$. This means the total search time is bounded by the time bound of executing a sequence of 4 attacks on the full dataset. Further, $T_r \leq \frac{4}{3} \times m \times N \times T_c$, which means the search time of an attack is bounded by $\frac{4}{3}$ of the allowed runtime to execute the attack.

# B  Search Space of $\mathbb{S} \times \mathbb{L}$

## B.1  Loss function space $\mathbb{L}$

Recall that the loss function search space is defined as:

```
 (Loss Function Search Space)
ℒ ::=   targeted Loss, n with Z |
        untargeted Loss with Z |
        targeted Loss, n - untargeted Loss with Z
Z ::=   logits | probs
```

To refer to different settings, we use the following notation:

- U: for the untargeted loss,
- T: for the targeted loss,
- D: for the targeted − untargeted loss
- L: for using logits, and
- P: for using probs

For example, we use DLR-U-L to denote untargeted DLR loss with logits. The loss space used in our evaluation is shown in Table 3. For hinge loss, we set $\kappa = -\infty$ in implementation to encourage stronger adversarial samples. Effectively, the search space includes all the possible combinations expect that the cross-entropy loss supports only probability. Note that although $\ell_{\text{DLR}}$ is designed

Table 3: Loss functions and their modifiers. ✓ means the loss supports the modifier. P means the loss always uses Probability.

| Name | Targeted | Logit/Prob | Loss |
|------|:--------:|:----------:|------|
| $\ell_{\texttt{CE}}$ | ✓ | P | $\ell_{\texttt{CrossEntropy}} = -\sum_{i=1}^{K} y_i \log(Z(x)_i)$ |
| $\ell_{\texttt{Hinge}}$ | ✓ | ✓ | $\ell_{\texttt{HingeLoss}} = \max(-Z(x)_y + \max_{i \neq y} Z(x)_i, -\kappa)$ |
| $\ell_{\texttt{L1}}$ | ✓ | ✓ | $\ell_{\texttt{L1}} = -Z(x)_y$ |
| $\ell_{\texttt{DLR}}$ | ✓ | ✓ | $\ell_{\texttt{DLR}} = -\dfrac{Z(x)_y - \max_{i \neq y} Z(x)_i}{Z(x)_{\pi_1} - Z(x)_{\pi_3}}$ |
| $\ell_{\texttt{LogitMatching}}$ | ✓ | ✓ | $\ell_{\texttt{LogitMatching}} = \|Z(x') - Z(x)\|_2^2$ |

Table 4: Generic parameters and loss support for each attack in the search space. For the `loss` column, "-" means the loss is from the library implementation, and ✓ means the attack supports all the loss functions defined in Table 3. In other columns ✓ means the attack supports all the values, and the attack supports only the indicated set of values otherwise.

| ATTACK | RANDOMIZE | EOT | REPEAT | LOSS | TARGETED | LOGIT/PROB |
|--------|-----------|-----|--------|:----:|:--------:|:----------:|
| FGSM | TRUE | $\mathbb{Z}[1, 200]$ | $^*\mathbb{Z}[1, 10000]$ | ✓ | ✓ | ✓ |
| PGD | TRUE | $\mathbb{Z}[1, 40]$ | $\mathbb{Z}[1, 10]$ | ✓ | ✓ | ✓ |
| DEEPFOOL | FALSE | 1 | 1 | ✓ | D | ✓ |
| APGD | TRUE | $\mathbb{Z}[1, 40]$ | $\mathbb{Z}[1, 10]$ | ✓ | ✓ | ✓ |
| C&W | FALSE | 1 | 1 | - | {U, T} | L |
| FAB | TRUE | 1 | $\mathbb{Z}[1, 10]$ | - | {U, T} | L |
| SQR | TRUE | 1 | $\mathbb{Z}[1, 3]$ | ✓ | ✓ | ✓ |
| NES | TRUE | 1 | 1 | ✓ | ✓ | ✓ |

for logits, and $\ell_{\texttt{LogitMatching}}$ is designed for targeted attacks, the search space still makes other possibilities an option (i.e., it is up to the search algorithm to learn which combinations are useful and which are not).

## B.2  Attack Algorithm & Parameters Space $\mathbb{S}$

Recall the attack space defined as:

$$\mathbb{S} ::= \quad \mathbb{S}; \mathbb{S} \mid \texttt{randomize}\ \mathbb{S} \mid \texttt{EOT}\ \mathbb{S}, \text{n} \mid \texttt{repeat}\ \mathbb{S}, \text{n} \mid$$
$$\texttt{try}\ \mathbb{S}\ \texttt{for}\ \text{n} \mid \text{Attack}\ \texttt{with}\ \text{params}\ \texttt{with}\ \text{loss} \in \mathbb{L}$$

`randomize`, `EOT`, `repeat` are the generic parameters, and `params` refers to attack specific parameters. The type of every parameter is either integer or float. An integer ranges from $p$ to $q$ inclusive is denoted as $\mathbb{Z}[p, q]$. A float range from $p$ to $q$ inclusive is denoted as $\mathbb{R}[p, q]$. Besides value range, prior is needed for parameter estimator model (TPE in our case), which is either uniform (default) or log uniform (denoted with $^*$). For example, $^*\mathbb{Z}[1, 100]$ means an integer value ranges from 1 to 100 with log uniform prior; $\mathbb{R}[0.1, 1]$ means a float value ranges from 0.1 to 1 with uniform prior.

Generic parameters and the supported loss for each attack algorithm are defined in Table 4. The algorithm returns a deterministic result if `randomize` is False, and otherwise the results might differ due to randomization. Randomness can come from either perturbing the initial input or randomness in the attack algorithm. Input perturbation is deterministic if the starting input is the original input or an input with fixed disturbance, and it is randomized if the starting input is chosen uniformly at random within the adversarial capability. For example, the first iteration of FAB uses the original input but the subsequent inputs are randomized (if the randomization is enabled). Attack algorithms like SQR, which is based on random search, has randomness in the algorithm itself. The deterministic version of such randomized algorithms is obtained by fixing the initial random seed.

The definition of `randomize` for FGSM, PGD, NES, APGD, FAB, DeepFool, C&W is whether to start from the original input or uniformly at random select a point within the adversarial capability. For SQR, random means whether to fix the seed. We generally set `randomize` to be True to allow repeating the

Table 5: List of attack specific parameters. The parameter names correspond to the names in the library implementation

| Attack | Parameter | Range and prior |
|---|---|---|
| NES | step | $\mathbb{Z}[20, 80]$ |
| | rel_stepsize | $^*\mathbb{R}[0.01, 0.1]$ |
| | n_samples | $\mathbb{Z}[400, 4000]$ |
| C&W | confidence | $\mathbb{R}[0, 0.1]$ |
| | max_iter | $\mathbb{Z}[20, 200]$ |
| | binary_search_steps | $\mathbb{Z}[5, 25]$ |
| | learning_rate | $^*\mathbb{R}[0.0001, 0.01]$ |
| | max_halving | $\mathbb{Z}[5, 15]$ |
| | max_doubling | $\mathbb{Z}[5, 15]$ |

| Attack | Parameter | Range and prior |
|---|---|---|
| PGD | step | $\mathbb{Z}[20, 200]$ |
| | rel_stepsize | $^*\mathbb{R}[1/1000, 1]$ |
| APGD | rho | $\mathbb{R}[0.5, 0.9]$ |
| | n_iter | $\mathbb{Z}[20, 500]$ |
| FAB | n_iter | $\mathbb{Z}[10, 200]$ |
| | eta | $\mathbb{R}[1, 1.2]$ |
| | beta | $\mathbb{R}[0.7, 1]$ |
| SQR | n_queries | $\mathbb{Z}[1000, 8000]$ |
| | p_init | $\mathbb{R}[0.5, 0.9]$ |

attacks for stronger attack strength, yet we set `DeepFool` and `C&W` to False as they are minimization attacks designed with the original inputs as the starting inputs.

The attack specific parameters are listed in Table 5, and the ranges are chosen to be representative by setting reasonable upper and lower bounds to include the default values of parameters. Note that `DeepFool` algorithm uses the loss D to take difference between the predictions of two classes by design (i.e., `targeted` − `untargeted` loss). FAB uses loss similar to `DeepFool`, and `C&W` uses the hinge loss. For `C&W` and `FAB`, we just take the library implementation of the loss (i.e. without our loss function space formulation).

### B.3  Search space conditioned on network property

Properties of network defenses (e.g. randomized, detector, obfuscation) can be used as a prior to reduce the search space. In our work, `EOT` is set to be 1 for deterministic networks. Using meta-learning techniques to reduce the search space is left for future work.

## C  Evaluation Metrics Details

We use the following $L_\infty$ criteria in the formulation:

MISCLASSIFICATION
$L_\infty$ ATTACK    $\|x' - x\|_\infty \leq \epsilon$  s.t.  $\hat{f}(x') \neq \hat{f}(x)$

We remove the misclassified clean input as a pre-processing step, such that the evaluation is performed only on the subset of correctly classified samples (i.e. $\hat{f}(x) = y$).

**Sequence of Attacks**    Sequence of attacks defined in Section 3 is a way to calculate the per-example worst-case evaluation, and the four attack ensemble in AutoAttack is equivalent to sequence of four attacks [APGD$_{CE}$, APGD$_{DLR}$, FAB, SQR]. Algorithm 2 elaborates how the sequence of attacks is evaluated. That is, the attacks are performed in the order they were defined and the first sample $x'$ that satisfies the criterion $c$ is returned.

**Robust Test Error (Rerr)**    Following Stutz et al. (2020), we use the *robust test error (Rerr)* metric to combine the evaluation of defenses with and without detectors. Rerr is defined as:

$$Rerr = \frac{\sum_{n=1}^{N} \max_{d(x',x) \leq \epsilon, g(x')=1} \mathbb{1}_{f(x') \neq y}}{\sum_{n=1}^{N} \max_{d(x',x) \leq \epsilon} \mathbb{1}_{g(x')=1}} \tag{5}$$

where $g \colon \mathbb{X} \to \{0, 1\}$ is a detector that accepts a sample if $g(x') = 1$, and $\mathbb{1}_{f(x') \neq y}$ evaluates to one if $x'$ causes a misprediction and to zero otherwise. The numerator counts the number of samples that are both accepted and lead to a successful attack (including cases where the original $x$ is incorrect), and the denominator counts the number of samples not rejected by the detector. A defense without a detector (i.e., $g(x') = 1$) reduces Eq. 5 to the standard Rerr. We define *robust accuracy* as $1-$ Rerr.

---
**Algorithm 2:** Sequence of attacks
---
  **def** `SeqAttack`
    **Input:** model $f$, data $x$, sequence attacks $\mathcal{S} \subseteq \mathbb{S}$, network transformation $t \in \mathbb{T}$,
          criterion function $c$
    **Output:** $x'$
**1**    **for** $\theta \in S$ **do**
**2**       $x' = a_{[\theta, t]}(x, f)$;
**3**       **if** $c(f, x', x)$ **then**
**4**           **return** $x'$

**5**    **return** $x'$
---

Note however that Rerr defined in Eq. 5 has intractable maximization problem in the denominator, so Eq. 6 is the empirical equation used to give an upper bound evaluation of Rerr. This empirical evaluation is the same as the evaluation in Stutz et al. (2020).

$$Rerr = \frac{\sum_{n=1}^{N} max\{\mathbb{1}_{f(\boldsymbol{x}_n) \neq y_n} g(\boldsymbol{x}_n), \mathbb{1}_{f(\boldsymbol{x}'_n) \neq y_n} g(\boldsymbol{x}'_n)\}}{\sum_{n=1}^{N} max\{g(\boldsymbol{x}_n), g(\boldsymbol{x}'_n)\}} \tag{6}$$

**Detectors** For a network $f$ with a detector $g$, the criterion function $c$ is misclassification with the detectors, and it is applied in line 3 in Algorithm 2. This formulation enables per-example worst-case evaluation for detector defenses.

Note that we use a zero knowledge detector model, so none of the attacks in the search space are aware of the detector. However, $\text{A}^3$ search adapts to the detector defense by choosing attacks with higher scores on the detector defense, which for `A4`, `C15` and `C24` does lead to lower robustness.

$$\begin{array}{l} \text{MISCLASSIFICATION} \\ \quad L_\infty \text{ ATTACK} \\ \text{WITH DETECTOR } g \end{array} \quad \|x' - x\|_\infty \leq \epsilon \quad \text{s.t.} \quad \begin{array}{l} \hat{f}(x') \neq \hat{f}(x) \\ g(x') = 1 \end{array}$$

**Randomized Defenses** If $f$ has randomized component, $f(x_n)$ in Eq. 6 means to draw a random sample from the distribution. In the evaluation metrics, we report the mean of adversarial samples evaluated 10 times using $f$.

## D   Discovered Adaptive Attacks

To provide more details on Table 1, Table 7 shows the network transformation result, and Table 6 shows the searched attacks and losses during the attack search.

**Network Transformation Related Defenses** In the benchmark, there are 4 defenses that are related to the network transformations. JPEG compression (`JPEG`) applies image compression algorithm to filter the adversarial disturbances and to make the network non-differentiable. Reverse sigmoid (`RS`) is a special layer applied on the model's logit output to obfuscate the gradient. Thermometer Encoding (`TE`) is an input encoding technique to shatter the linearity of inputs. Random rotation (`RR`) is in the family of randomized defense which rotates the input image by a random degree each time. Table 7 shows where the defenses appear and what network processing strategies are applied.

**Diversity of Attacks** From table 6, the majority of attack algorithms searched are `APGD`, which shows the attack is indeed a strong attack. The second or third attack can be a non-effective weak attack like `FGSM` and `DeepFool` in some cases, and the reason is that the noise in the untargeted CE loss tie-breaker determines the choice of attack when none of the samples are broken by the searched attacks. In these cases, the arbitrary choice is acceptable as none of the other attacks are effective. The loss functions show variety, yet `Hinge` and `DLR` appear more often than CE even we use CE loss as the tie-breaker. This challenges the common practise of using CE as the loss function by default to evaluate adversarial robustness.

Table 6: Time limit (TL), attacks and losses result. Due to the cost of A10, only one attack is searched and used. The Loss follows the format: **Loss - Targeted - Logit/Prob**. The abbreviations are defined in Section B.

| | TL(s) | Attack1 | Loss1 | Attack2 | Loss2 | Attack3 | Loss3 |
|---|---|---|---|---|---|---|---|
| A1 | 0.5 | APGD | Hinge-T-P | APGD | L1-D-P | APGD | CE-T-P |
| A2 | 0.5 | APGD | Hinge-U-L | APGD | DLR-T-L | APGD | CE-D-P |
| A3 | 0.5 | APGD | CE-T-P | APGD | DLR-U-L | APGD | L1-T-P |
| A4 | 0.5 | FAB | −F-L | APGD | LM-U-P | DeepFool | DLR-D-L |
| A5 | 0.5 | APGD | Hinge-U-P | APGD | Hinge-U-P | PGD | DLR-T-P |
| A6 | 0.5 | APGD | L1-D-L | APGD | DLR-U-L | APGD | Hinge-T-L |
| A7 | 0.5 | APGD | DLR-T-P | APGD | DLR-U-L | APGD | Hinge-T-L |
| A8 | 1 | APGD | L1-U-P | APGD | CE-U-P | APGD | CE-D-P |
| A9 | 1 | APGD | DLR-U-L | APGD | Hinge-U-P | APGD | CE-U-L |
| A9' | 30 | NES | Hinge-U-P | - | - | - | - |
| B10 | 3 | APGD | DLR-U-L | SQR | DLR-U-S | DeepFool | CE-D-P |
| B11 | 3 | APGD | Hinge-T-P | DeepFool | L1-D-L | PGD | CE-D-P |
| B12 | 3 | APGD | Hinge-T-P | DeepFool | Hinge-D-P | DeepFool | L1-D-L |
| B13 | 3 | APGD | CE-D-L | APGD | DLR-F-P | DeepFool | CE-D-L |
| B14 | 3 | APGD | Hinge-T-L | APGD | CE-U-P | C&W | −U-L |
| C15 | 2 | SQR | DLR-U-L | SQR | DLR-T-L | APGD | Hinge-U-P |
| C16 | 3 | FAB | −F-L | APGD | L1-T-L | FAB | −F-L |
| C17 | 3 | APGD | L1-D-P | APGD | CE-F-P | APGD | DLR-T-L |
| C18 | 3 | SQR | Hinge-U-L | SQR | L1-U-L | SQR | CE-U-L |
| C19 | 3 | APGD | L1-D-P | C&W | Hinge-U-L | PGD | Hinge-T-L |
| C20 | 3 | APGD | Hinge-U-L | APGD | DLR-T-L | FGSM | CE-U-P |
| C21 | 3 | APGD | Hinge-U-L | APGD | DLR-T-L | FGSM | DLR-U-P |
| C22 | 3 | PGD | DLR-U-P | FGSM | L1-U-P | FGSM | DLR-U-L |
| C23 | 3 | APGD | L1-T-L | PGD | L1-U-P | PGD | L1-U-P |
| C24 | 2 | PGD | L1-T-P | APGD | CE-T-P | APGD | L1-U-L |

Table 7: List of network processing strategy used on relevant benchmarks. The format is **defense-policy**. The defenses are defined in Section D. For layer removal policies, 1 means to remove the layer, 0 means not to remove the layer. For BPDA policies, I means identity, and C means using the network with two convolutions having ReLU activation in between.

| | Removal Policies | BPDA Policies |
|---|---|---|
| A2 | - | TE-C |
| A3 | JPEG-1 RS-1 | JPEG-I |
| A4 | RR-0 | - |
| A6 | JPEG-1 RS-1 RR-1 | TE-C, JPEG-I |
| C20 | JPEG-0 RS-0 | JPEG-I |
| C21 | JPEG-1 RS-1 | JPEG-I |
| C22 | RR-0 | - |
| C23 | RR-0 | - |

# E   Scalability Study

Here we provide details on scalability study in Section 5.

We designed an extended search space with addition of 8 random attacks and 4 random losses to test the scalability of $A^3$. Random attack is to sample a point inside of the disturbance budget uniformly at random, and random loss is $\ell_{CE}$ with random sign. In our original search space for a single attack, the number of attacks is 8 and the number of losses is 4 ($8 \times 4$), so the extended search space ($16 \times 8$) has $4\times$ the search space compared with the original space. In the other setting, we use half of the samples ($n = 50$) to check $A^3$ performance with halved search time. We evaluate block A models except A9 model because of the high variance in result (around $\pm 1.5$) due to the obfuscated nature of the defense.

Table 8: Evaluating scalability of $A^3$. Original search space corresponds to the search space defined in Appendix B. Extended search space additionally contains 8 random attacks and 4 losses.

| Net | AA | Original Search Space | | Extended Search Space | |
|-----|-----|-----|-----|-----|-----|
| | | Normal | n=50 | k=64 | k=96 |
| A1 | 44.78 | 44.69 | 44.93 | 44.80 | 44.80 |
| A2 | 2.29 | 1.96 | 2.09 | 2.14 | 1.83 |
| A3 | 0.59 | 0.11 | 0.11 | 0.11 | 0.10 |
| A4 | 6.17 | 3.04 | 3.15 | 3.47 | 2.89 |
| A5 | 22.30 | 12.14 | 12.53 | 11.65 | 11.85 |
| A6 | 4.14 | 3.94 | 3.86 | 4.43 | 4.43 |
| A7 | 2.85 | 2.71 | 2.78 | 2.79 | 2.76 |
| A8 | 19.82 | 11.11 | 11.52 | 13.02 | 11.09 |
| Avg | 12.87 | 9.96 | 10.12 | 10.30 | 9.97 |

We show the result in Table 8. We see a minor drop in performance with the extended search space or with half of the samples, and $A^3$ still gives competitive evaluation in these scenarios. When increasing the number of trials to 96 on the scaled dataset, the result reaches same performance.

The redundancy of $m = 3$ attack is an explanation of $A^3$ giving competitive performance in these scenarios. As long as one strong attack is found within the 3 attacks, the robustness evaluation is competitive.

## F  Ablation Study

Here we provide details on the ablation study in Section 5.

### F.1  Attack Algorithm & Parameters

In the experiment setup, the search space includes four attacks (FGSM, PGD, DeepFool, C&W) with their generic and specific parameters shown in Table 4 and Table 5 respectively. The loss search space is fixed to the loss in the original library implementation, and the network transformation space contains only BPDA. *Robust accuracy* (Racc) is used as the evaluation metric. The best Racc scores among FGSM, PGD, DeepFool, C&W with library default parameters are calculated, and they are compared with the Racc from the attack found by $A^3$.

The result in Table 9 shows the average robustness improvement is 5.5%, up to 17.3%. PGD evaluation can be much stronger after tuning by $A^3$, reflecting the fact that insufficient parameter tuning in PGD was a common cause to over-estimate the robustness in literature. At closer inspection, the searched attacks have larger step sizes (typically 0.1 compared with 1/40), and higher number of attack steps (60+ compared with 40).

### F.2  Loss

Figure 2 shows the comparison between TPE with loss formulation and TPE with default loss. The search space with default loss means the space containing only L1 and CE loss, with only untargeted loss and logit output. The result shows the loss formulation gives 3.0% improvement over the final score.

### F.3  TPE algorithm vs Random

In this experiment, we take $n = 100$ samples uniformly at random and run both TPE and random search algorithm on block A models. We record the progression of the best score in $k = 100$ trials. We repeat the experiment 5 times and average across the models and repeats to obtain the progression graph shown in Figure 2. The result shows that TPE finds better scores by an average of 1.3% and up to 8.0% (A6).

Table 9: Comparison with library default parameters and the searched best attack. The implementations of `FGSM`, `PGD`, and `DeepFool` are based on `FoolBox` (Rauber et al., 2017) version 3.0.0, `C&W` is based on `ART` (Nicolae et al., 2018) version 1.3.0.

| | Library Impl. | | | $\mathtt{A}^3$ | | |
|---|---|---|---|---|---|---|
| Net | Racc | Attack | | Racc | Δ | Attack |
| A1 | 47.1 | C&W | | 47.0 | -0.1 | PGD |
| A2 | 13.4 | PGD | | 6.7 | -6.8 | PGD |
| A3 | 35.9 | DeepFool | | 30.3 | -5.6 | PGD |
| A4 | 6.6 | DeepFool | | 6.6 | 0.0 | DeepFool |
| A5 | 14.5 | PGD | | 8.4 | -6.1 | PGD |
| A6 | 35.0 | PGD | | 17.3 | -17.7 | PGD |
| A7 | 6.9 | C&W | | 6.6 | -0.3 | C&W |
| A8 | 25.4 | PGD | | 14.7 | -10.7 | PGD |
| A9 | 64.7 | FGSM | | 62.4 | -2.3 | PGD |

In practice, random search algorithm is simpler and parallelizable. We observe that random search can achieve competitive performance as TPE search.

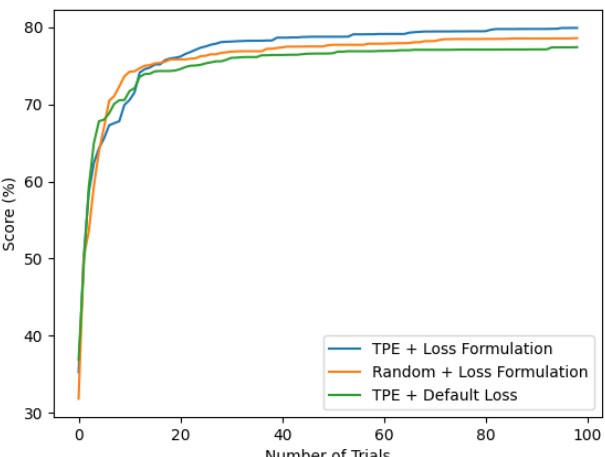

Figure 2: The best score progression measured by the average of 5 runs of models `A1` to `A9`.

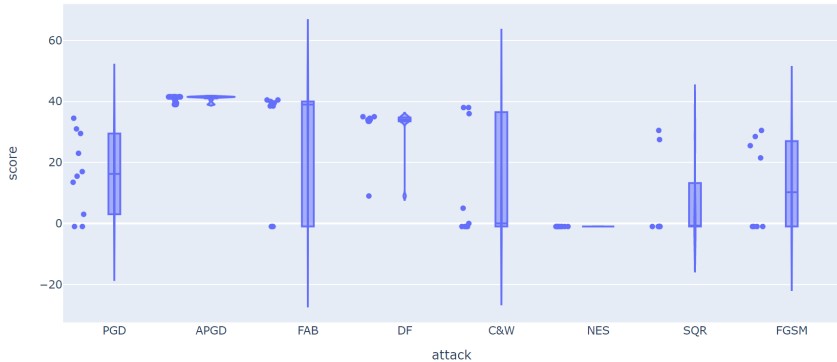

Figure 3: Attack-score distribution generated by `TPE` algorithm on `A1` model. Scores with negative values corresponds to the time-out trials.

Table 10: Three independent runs and confidence intervals of $A^3$ for models in Block A and B. The bold numbers show the worst case evaluation for each model. Each confidence interval is calculated as the plus and minus the standard deviation value across the three runs. Note, that the numbers from run 3 are identical to the numbers reported in Table 1.

| | Run | | | $\sigma$ |
|---|---|---|---|---|
| Net | 1 | 2 | 3 | Confidence Interval |
| A1 | 44.79 | 44.7 | **44.69** | $44.73 \pm 0.04$ |
| A2 | 2.23 | 2.13 | **1.96** | $2.11 \pm 0.11$ |
| A3 | **0.10** | 0.10 | 0.11 | $0.10 \pm 0.01$ |
| A4 | **3.00** | 3.32 | 3.04 | $3.12 \pm 0.14$ |
| A5 | 12.73 | 12.74 | **12.14** | $12.54 \pm 0.28$ |
| A6 | 4.18 | 4.11 | **3.94** | $4.08 \pm 0.10$ |
| A7 | 2.73 | **2.71** | 2.71 | $2.72 \pm 0.01$ |
| A8 | 10.86 | **10.49** | 11.11 | $10.82 \pm 0.25$ |
| A9 | 62.62 | **62.31** | 63.56 | $62.83 \pm 0.53$ |
| B10 | 62.80 | 62.83 | **62.79** | $62.81 \pm 0.02$ |
| B11 | 60.43 | 60.04 | **60.01** | $60.16 \pm 0.19$ |
| B12 | **59.54** | 59.54 | 59.56 | $59.55 \pm 0.01$ |
| B13 | **59.22** | 59.32 | 59.51 | $59.35 \pm 0.12$ |
| B14 | **57.11** | 57.24 | 57.16 | $57.17 \pm 0.05$ |

## G  Attack-Score Distribution during Search

The analysis of attack-score distribution is useful to understand $A^3$. Figure 3 shows the distribution when running $A^3$ on network A1. In this experiment, the number of trials is $k = 100$ and the initial dataset size is $n = 200$, the time budget is $T_c = 0.5$, and we use the search space defined in Appendix B. We used single GTX1060 on this experiment. We can observe the following:

- The expensive attack times out when $T_c$ values are small. Here the expensive attack NES gets time-out because a small $T_c$ is used.

- The range and prior of attack parameters can affect the search. As we see cheap FGSM gets time-out because the search space includes large repeat parameter.

- Different attack algorithms have different parameter sensitivity. For examples, PGD has a large variance of scores, but APGD is very stable.

- TPE algorithm samples more attack algorithms with high scores. Here, there are 18 APGD trials and only 7 NES trials. TPE favours promising attack configurations so that better attack parameters can be selected during the SHA stage.

- The top attacks have similar performance, which means the searched attack should have low variance in attack strength. In practice, the variance among the best searched attacks is typically small ($\pm 0.2\%$).

## H  Analysis of $A^3$ Confidence Interval

We evaluated $A^3$ using three independent runs for models in Block A and B as reported in Table 10. The result shows typically small variation across different runs (typically less than $\pm 0.2\%$), which means $A^3$ is consistent for robustness evaluation.

Confidence varies across different models, and the typical reason is the variance of the attacks on the same model. For examples, models A8, A9 are obfuscated and A5 is randomized, the attack has large variance due to the nature of these defenses.