# OpenReview forum: "Automated Discovery of Adaptive Attacks on Adversarial Defenses"
_NeurIPS.cc/2021/Conference — NeurIPS 2021 Poster_

### Official Review · Reviewer_UerA · 2021-07-11

**Rating:** 6
**Confidence:** 4

**Summary:**

The paper advances state-of-the-art automated pen-testing tools for breaking defenses designed to protect deep neural networks against norm-based adversarial attacks as a search problem. The search process first considers techniques to simplify the network being attacked (called Network Transformations), followed by a search over the space of existing attacks (and common ways to modify these attacks). Given the search space is large, the authors try to improve the efficiency by leveraging sub-sampling methods.

**Ethical Concerns:**

See limitations and societal impacts.

**Ethics Review Area:**

["Inappropriate Potential Applications & Impact  (e.g., human rights concerns)"]

**Limitations And Societal Impact:**

The idea of developing automated pen-testing tools is often discussed in the context of correctly calibrating the security offered by existing defenses. At the same time, it benefits an attacker, who now has to spend less resources if figuring out the best attack against a defense.

While this paper is not solely responsible in this regard, we should consider a better way to release such methods (say, learn from the best practices in cybersecurity). In cybersecurity, vulnerabilities often found by automated pen-testing software have know fixes/patches; this is not presently the case for deep neural networks (esp. if we consider real-world attacks and defenses). Thus, we (as a community) need to devote more thought before arming attackers with such automated software, which must consider re-evaluating research progress in this direction starting from the impact of releasing code to accepting such papers etc.

**Main Review:**

### Strengths

- The paper is easy to follow.
- The paper structures the idea of adaptive attacks into a search algorithm, making existing tricks leveraged in attacks simply a search node in the space of attacks.
- The results show that, in certain cases, their improvement to AutoAttack can result is finding stronger or faster attacks. I appreciate that they to do a fair evaluation, they also empower AutoAttack with some additions (such as BPDA) that their framework leverages.
- The analysis of how the attacks found relate to existing observations made by existing research was an intersting read.
- The ablation studies does do help to ensure that various parts of the search algorithm do indeed help in finding the best attack.

### Weaknesses / Correctness / Clarification

- The paper adds a bunch of bells-and-wistles to AutoAttack and is more of an automated pen-testing software paper; I could not find much novelty or insights for improving the attack of deep neural networks. While auto attack proposes two extensions of PGD and then considers an ensemble approach, the contribution of this paper seems to be a search process formulated over the space of existing attacks (modifications thereof), and network transformations etc.
- $A^3$ hardly provides improvement over adversarial training based defenses. This indicates that without novel attacks (lines 233-235), the search procedure may not be add any value in finding the best attack against SOTA defenses, tying back to the lack of novelty pointed above.
- While reading, I kept thinking if a meta-learning approach to the search would be an interesting consideration. I found the authors mentioned this as future work at the very end. I surely think using something similar to meta-learning would add more novelty that simply bucketing existing tips and tricks at different layers of a computationally expensive search process.
- Why did the authors evaluate on a sub-set of the defenses on which AutoAttack was evaluated? It sort of gives the impression that $A^3$ did not provide any gains compared to AA (w.r.t accuracy or time) on the defenses omitted.

**Needs Ethics Review:**

Yes

**Time Spent Reviewing:**

5-6

---

> ### Author Response · Authors · 2021-08-10
> **Author Response**
>
> Thank your for the feedback and questions, answers to which we summarized below:
>
> > Why does A^3 provide small improvement over adversarial training based defenses? What value does A^3 provide in this case?
>
> Currently, the main benefit of A^3 is evaluating new defences that include additional components designed to hinder the attacker. In this case, providing adaptive evaluation is critical and our tool can improve the evaluation by up to 50%.
>
> For defences based on adversarial training, existing attacks already provide good robustness estimates. In this case, the main practical value of our tool is in improved runtime.
>
> > The paper is more of an automated pen-testing software paper
>
> The automated pen-testing software may be a fitting description to this work -- as a step towards making the process of evaluating defences systematic and reliable.
>
> Most importantly, we showed that such an approach can significantly improve the evaluation of recent state-of-the-art adversarial defences, making it a useful tool for the community as a whole.
>
> > While auto attack proposes two extensions of PGD and then considers an ensemble approach, the contribution of this paper seems to be a search process formulated over the space of existing attacks (modifications thereof), and network transformations etc.
>
> Correct, but this is exactly what we believe is needed and maybe we should explain it better in our paper. That is, our goal was to:
> - Design a tool that learns from and generalizes the existing N techniques,
> - Rather than proposing N + 1 technique
>
> In other words, instead of:
> - proposing 1 or 2 extensions of existing attacks as suggested by the reviewer (and as done by AutoAttack),
> - we proposed a _search space of exponentially many extensions_ that can be combined in different ways. Given that this search space consists of existing attacks, each of them is indeed not novel.
>
> > Why did the authors evaluate on a sub-set of the defenses on which AutoAttack was evaluated?
>
> We diversified our evaluation as follows:
> - Most deterministic CIFAR10 epsilon=8/255 models (22 out of 25) from AutoAttack paper are variants of adversarial training or regularization on the loss surface. We chose 5 of the top 10 models from robustbench to represent this school of defense.
> - We include all randomized defenses from AutoAttack except for ME-net, which was too expensive to evaluate due to its matrix factorization component.
>
> In addition to the subset of models from AA and robustbench, we included a number of recent papers that use:
> - gradient obfuscated defences (A3, A8, C20, C21), and
> - detectors defenses (A4, C15, C24)

---

> > ### Comment · Reviewer_UerA · 2021-08-31
> > **Thanks for the response; still not convinced of the novelty.**
> >
> > I would like to thank the authors for their response. While some of the clarifications raised were answered, I see that the authors mostly agree with my characterization of their work; simply stating that the aim of the paper was to elucidate the search space and (exponentially) increase attack options that tie together existing attacks doesn't really convince me of the novelty (again, I already understood this). Having said that, I did initially consider the paper to be marginally above threshold and will stick to that rating.

---

### Official Review · Reviewer_Z4Ru · 2021-07-12

**Rating:** 5
**Confidence:** 5

**Summary:**

This paper's main premise is that existing defense evaluations (whether adaptive or based on an ensemble of fixed attacks) only rely on a small number of common techniques and that a search over the space of possible attacks can efficiently uncover strong attacks.

**Limitations And Societal Impact:**

As stated in the main review, the generality of the approach should be further discussed, as well possible limitations against future defenses.

**Main Review:**

The automatization of defense evaluations is ultimately a good thing, I think, if it leads to fewer clear mistakes and omissions in these evaluations. The proposal in this paper is quite natural and straightforward, in that it combines existing techniques and evaluates them against previously-evaluated defenses. To make this contribution more compelling, the paper could benefit from expanding along the following two axes:

1) Generality

Existing attacks indeed seem to make use of only a small number of different components. This is especially true for defenses that don't do anything "weird" at evaluation time. And many of the defenses (re-)evaluated in this paper are of this form (e.g., all forms of adversarial training).

But now let's imagine that a new defense comes out, that hasn't been independently evaluated before.
Should we trust the procedure outlined in this paper to find a strong attack?
I encourage the authors to explicitly think about this next step: what would it take to design a defense that is (a) non-robust, yet (b) resits your automated attack procedure?
My guess is that it wouldn't be too hard to build such a defense, and that this should be acknowledged.

The paper's claim that it automates the manual search for attacks (e.g., of Athalye et al. or Tramer et al.) is also hard to assess, as the proposed attack search is only evaluated against a small subset of the defenses evaluated in those works. Unless I am mistaken, I count 3/9 defenses in common with the work of Athalye et al. and 3/13 in common with the work of Tramer et al.

It thus seems challenging to reach a conclusion on whether automated search techniques could have found good attacks for some of these challenging defenses. And as noted above, we also don't know how well the proposed search procedure works against defenses that haven't been re-evaluated before.

2) Ease-of-use

Related to the above discussion on the attack's generality is the question of its ease-of-use. My worry here is that the more general an automated attack aims to become, the harder it will become to use as it will inevitably require the defense to be represented in a tool-specific way.
E.g., certain defenses may not be designed in a way that makes it easy to replace individual components with a BPDA approximation. Or some defenses (e.g., detectors) may produce outputs that cannot immediately be converted into the type of score your attack expects.
Is there reason to believe that this (manual) process of adapting a defense to an automated attack tool is necessarily easier and less error-prone than the implementation of an appropriate attack?


Additional comments:

- how do you automate the process of applying EOT? Often, different defenses will rely on very different forms of random transformations.
- For the comparisons in Table 1, it would be worth including some confidence intervals when comparing A3 and AA. E.g., for all the B* models, the difference between the two attacks is most certainly not statistically significant.
- Why do you include attacks such as FGSM or DeepFool in your search space, when these are known to super-seeded by newer stronger attacks?

**Time Spent Reviewing:**

2

---

> ### Author Response · Authors · 2021-08-10
> **Author Response**
>
> We thank the reviewer for the insightful review and include our response below:
>
> > Should we trust the procedure outlined in this paper to find a strong attack?
>
> To obtain provable guarantees one would need to use the line of work that develops certified robustness evaluation (i.e, based on convex relaxations or interval bound propagation). Unfortunately, such techniques have their own limitations as they scale only to small networks and introduce approximation errors (which scale exponentially with the network size).
>
> What our approach can however provide is to extend the empirical robustness evaluation from a single attack (or small set of attacks) to a large search space that is explored automatically (i.e., gives a lower bound evaluation over the search space). That is, as long as a strong attack is included in the search space, we will eventually find it.
>
> > What would it take to design a defense that is (a) non-robust, yet (b) resits A^3’s automated attack procedure?
>
> Indeed, this is possible and has been at the core of the recent game between attacks and defences. What tools like AA and ours provide is that instead of overfitting to a single attack, the defence has to do a much better job at improving over a gradually expanding set of defenses. This makes the whole process much more systematic and reliable.
>
> > How well the proposed search procedure works against defenses that haven't been re-evaluated before?
>
> We included a small study along these lines in our evaluation. In particular, we showed three cases C15,C18,C24 (lines 249 - 273) where strong attacks are found without the handcrafted attacks in the search space.
>
> > Does the paper's claim that it automates the manual search for attacks (e.g., of Athalye et al. or Tramer et al.)?
>
> Our goal was not to claim that we fully automate the process, see our limitations in the conclusion section (lines 342+). Our goal was to make a step in this direction by showing how the search space can be designed, what improvement it can provide, and open-sourcing our work such that it can be further extended.
>
> > My worry here is that the more general an automated attack aims to become, the harder it will become to use as it will inevitably require the defense to be represented in a tool-specific way.
>
> This is a very interesting point. Indeed, applying any tool (not just ours) to a large number of models is very challenging from the engineering perspective (especially when the models are coming out from the latest research). For example, we have spent significant effort in making it possible to include a variety of models in our evaluation and this is where initiatives such as robustbench are extremely valuable.
>
> The good part is that once the model is in a suitable form (i.e., a computational graph, say from tensorflow or onnx), then the analysis generalizes well. Currently, most of the manual effort goes into getting to the point that the model can be run and computational graphs can be extracted. This is especially true when it comes to supporting multiple frameworks, each of which handles the models in a different way, external API, etc.
>
> > How do you automate the process of applying EOT?
>
> EOT is applied independently of the transformation that is being applied. For BPDA, we include different ways how the layers can be approximated (Table 2).
>
> > Can you include confidence intervals when comparing A3 and AA?
>
> Yes, we are happy to include those in the revised version of our paper. We note however that for B* models, the AutoAttack performance is reported as the best among different AutoAttack runs. This improves by ~0.05% compared to using a single AutoAttack run (for our method we report only a single evaluation).
>
> > Why do you include attacks such as FGSM or DeepFool in your search space, when these are known to super-seeded by newer stronger attacks?
>
> There are two main reasons:
>
> - First, weak defenses can be surprisingly effective in special cases. For example, in defense C24, the attack PGD on L1 loss, which is considered a weak attack, fools the detector because of its weakness.
>
> - Second, we are not using any prior when designing the space search. In particular, whenever a new attack is designed it can simply be added to the search space. Then, the goal of the search algorithm is to be powerful enough to perform the search efficiently (we have included an ablation study to investigate this point).
>
> Essentially, we want to avoid making any assumptions of what is useful or not and let the search algorithm learn automatically.

---

### Official Review · Reviewer_rcDN · 2021-07-16

**Rating:** 4
**Confidence:** 4

**Summary:**

This paper claim that adaptive attacks are composed from a set of reusable building blocks that can be formalized in a search space and used to automatically discover attacks for unknown defenses. Based on this observation, they propose an automated framework of discovering adaptive attacks, which does not rely on the expert's knowledge priori.

**Limitations And Societal Impact:**

The authors have addressed the limitations and potential negative societal impact of their work.

**Main Review:**

**Originality:**
The work is meaningful in engineering, but its contribution and novelty are limited.

Q1: There is a problem with the motivation of this paper, that is, this work believes that adaptive attacks in Tramer et al. (2020) require expert knowledge, so A^3 is proposed. But A^3 does not have experiments to prove that it is superior to an adaptive attack based on expert knowledge. Pang et al. (2019) and Xiao et al. (2020) in Tramer et al. (2020) are both close to 0% after adaptive attacks, and A^3 cannot do this. So, what is the goal of this paper? Perhaps the strongest adaptive attack still requires experts to design.

Q2: Composite Adversarial Attack (CAA) [1] is proposed for automatically searching the best combination of attack algorithms and their hyper-parameters from a candidate pool of 32 base attackers. The experimental result shows CAA beats 10 top attackers on 11 diverse defenses with a less elapsed time (6 × faster than AutoAttack) and achieves the new state-of-the-art on l∞, l2, and unrestricted adversarial attacks. Obviously, CAA and this work have similar ideas, and both are to evaluate the robustness of the model, but this work is not compared with CAA, nor does it cite CAA, even if CAA’s code is open.

**Quality:**
The experiment in this paper proves that it is better than AutoAttack, but there are several problems:

Q3: AutoAttack is a parameter-free, computationally affordable, and user-independent ensemble of attacks to test adversarial robustness and not specifically designed for adaptive attacks. From this perspective, if this work wants to highlight the advantages of adaptive attacks, the comparison with AutoAttack is unfair, even though AutoAttack has been improved with adaptive attacks. A better way is to explicitly compare with existing or adaptive attack methods in the original paper.

Q4: A^3 is closely related to the recent advances in AutoML, such as in the domain of neural architecture search (NAS). How to reflect the advantages of your search strategy? What’s more, the combination of NAS and adversarial attacks already exists (CAA[1]) and should be compared with it.

Q5: In the body, the experiment only has the setting of L∞, can A^3 be extended to other L-norm settings? There are other L-norm experiments in AutoAttack and CAA[1].


**Clarity:**
The paper is clear on the whole.

**Significance:**
This work is meaningful in engineering applications. If it is superior to adaptive attacks based on expert knowledge, it is vital for reliable evaluation of adversarial defenses.


[1] Xiaofeng Mao, Yuefeng Chen, Shuhui Wang, Hang Su, Yuan He, Hui Xue: Composite Adversarial Attacks. AAAI 2021: 8884-8892

**Time Spent Reviewing:**

20

---

> ### Author Response · Authors · 2021-08-10
> **Author Response**
>
> Thank you for the feedback, please find answers to your questions and main concerns below:
>
> > Can you please clarify the goal of this paper?
>
> The goal of our paper is to automate the tedious and time-consuming trial-and-error steps that domain experts perform manually today, allowing them to focus on the creative task of designing new attacks. That is, we can automate some part of the process of finding adaptive attacks, but not necessarily the full process. This is natural as finding truly new attacks is a highly creative process that is currently out of reach for fully automated techniques.
>
> > Perhaps the strongest adaptive attack still requires experts to design?
>
> Correct, we do not claim that we can outperform or completely replace a human expert in designing adaptive attacks. Instead, we show that we can formalize part of the process of finding adaptive attack and use it to successfully improve evaluation across a range of recent adversarial defences.
>
> However, we do note in some cases, our approach can also discover stronger attacks than a human expert (for example C15).
>
> > Can you compare A^3 to CAA?
>
> Thanks for pointing out the concurrent work of CAA, we are happy to include a comparison in the revised version of our work. In short, our tool improves upon CAA both in terms of the technical approach as well as experimental results. Concretely:
>
> *Techniques*:
> - CAA: the main idea is that individual attacks can be composed to achieve better performance. In contrast,
> - A^3 (our work): the main idea is that *the way adaptive attacks are designed today* can be formalized as a search space that includes three key components: network transformation, attack parameters, and loss functions. This is critical as it defines a much larger search space and allows reuse of low level components, rather than composing only the high level attacks.
>
> *Evaluation*:
> - CAA: is evaluated in 11 defenses and is better than AA by ~0.2% in 5 cases and by up ~1.0 - 2.0% in another 5 cases (Tables 2 and 3 in CAA)
> - A^3: we evaluated our work on 24 models. Similar to CAA, we can improve by up to 2.0% in 14 cases. However, unlike CAA, in the remaining 10 cases we further improve by 3.0% - 50.0%, significantly outperforming AA.
>
> > Can A^3 be extended to other L-norm settings?
>
> Yes, this can be done easily in the same way as CAA which includes adversarial attacks handling different Lp norms.
>
> > Can you compare with existing or adaptive attack methods in the original paper, instead of AA?
>
> We included exactly this comparison for some of the attacks in the text at lines 249-273. Here, we show that A^3 can "rediscover" attacks with similar strength or even improve upon the original adaptive attack (A^3 discovers attack with 26.87% robust accuracy compared to the 31.6% reported by the authors for C15).
>
> > How to reflect the advantages of AutoML and NAS in your search strategy?
>
> Indeed, there is a tight connection to both AutoML and NAS when it comes to designing the search algorithm. In our case, both the tree parzen estimation (TPE) and successive halving (SHA) are also techniques used in AutoML / NAS.
>
> > Could other algorithms improve the scalability even further?
>
> Yes, it is only natural that new advances in AutoML and NAS could also be applied to improve the performance of our search algorithm. Searching the best attack in our search space is a well defined optimization problem for which large number of methods can be applied.
>
> Our aim was to design a search algorithm that is efficient and scales to the large search space considered in our work (especially as the search space is expected to increase in the future). For this, we also provide practical bounds on the time budget and the worst-case search time (discussed at lines 190-195 and Appendix A).

---

> > ### Comment · Reviewer_rcDN · 2021-09-02
> > **Post-rebuttal thoughts**
> >
> > I have read the author's reply and other reviewer's comments. A^3 is a great framework for discovering adaptive attacks but the contribution is limited. CAA put forward similar ideas before A^3, but A^3 has not been cited and compared with CAA in practice, including different L-norm settings. I prefer to see the difference between the two under the same setting, so I keep the existing rating.

---

### Official Review · Reviewer_KocB · 2021-07-16

**Rating:** 5
**Confidence:** 4

**Summary:**

One major challenge in the research on adversarial attacks is the absence of methods that can reliably evaluate an adversarial defense. One the one hand, many defense approaches end up targetting particular attack methods. On the other hand, some promising defense approaches are viewed skeptically due to possible new attacks that could break them even if none currently exists. This paper addresses this critical challenge of developing adaptive adversarial attacks by combining some basic building blocks based on a particular defense. The presented approach is shown to outperform AutoAttack - a popularly used baseline attack by community developing adversarial defenses.

The submission presents an automated approach to the discovery of adversarial attacks (along the lines of AutoAttack) by exploring the search space of adversarial attacks formed by varying three components: (i) parameterized attack algorithms, (ii) network transformations, and (iii) loss functions.

Overall, the paper is an interesting improvement over the state-of-the-art and clarification to some questions below will help the reviewer better appreciate its overall contribution.

**Limitations And Societal Impact:**

Weaknesses

(-) The idea of replacing a human expert attacking a model with an algorithm is interesting. However, it is not clear if an automated approach can be as creative as a human expert. For example, layer removal when the inputs and outputs have the same shape (lines 128-129) can be easily circumvented by a human expert by adding redundant inputs or outputs that this automated tool may not discover. Can the network transformation take care of such cases?

(-) The experiment results on the top models (A1 and B10-B14) in Table 1 do not show very significant improvement over AutoAttack. Improvements against Madry et al (A1) is 0.9 and against robust models (B10-B14) is below 0.1. Were any ablation studies performed on models (B10-B14)?

(-)  The grammar presented here to represent the space of possible attacks  is quite simple. Would the defense algorithms not be able to exploit the knowledge of this grammar?

**Main Review:**

Strengths

+ The use of parameter search algorithms to tune adversarial attacks generated by a formal grammar is likely to be useful. More discussion on the parameter search and its efficacy/challenges may be helpful.

+ The paper includes a meticulous comparison with state-of-the-art. In particular, it highlights the key distinctions from AutoAttack and another recently proposed adaptive attack (Tramer et. al. 2020).

+ The paper has identified dimensions of different adaptive attack methods (attack algorithm params, network transformations,  loss functions), which could be possibly reused to develop new adaptive attack techniques.

+ The implementation is available open-source.



**Time Spent Reviewing:**

3 hours

---

> ### Author Response · Authors · 2021-08-10
> **Author Response**
>
> Thank you for the feedback and questions.
>
> > Can an automated approach be as creative as a human expert?
>
> While it is possible that the search discovers interesting combinations automatically (as illustrated at 249-273), we expect ours and similar search based approaches to assist experts rather than to replace them.
>
> In particular, the main goal of our approach is to automate the tedious and time-consuming trial-and-error steps that domain experts perform manually today, allowing them to focus on the creative task of designing new attacks (lines 335-336).
>
> > Can the network transformation handle redundant inputs or outputs?
>
> Yes, this can be easily handled by adapting the search space.
>
> > Could you provide ablation studies performed on models (B10-B14)? Why is there not a significant improvement for these models?
>
> We are happy to include additional discussion in the revised version of our paper.
>
> In short, these models aim to improve the adversarial training procedure rather than developing a new defence. For example, they employ additional unlabelled data for training, extensive hyperparameter tuning, instance weighting or loss regularization. As a result, existing adversarial attacks already provide close to optimal performance.
>
> This is in contrast to models that do design various types of new defences, evaluating which requires discovering a new adaptive attack. For these new defences evaluation is much more difficult and this is where our approach also improves the most.
>
> > Would the defense algorithms not be able to exploit the knowledge of this grammar?
>
> Yes, this is indeed the case and will be the case for any empirical attack that is available to the defence at design time (even if it comes from a very large grammar). The main insight of our work is that we can:
> - Identify a set of reusable components that can represent existing adaptive attacks, and
> - An extendable search space that can be enlarged with new components in the future.
>
> By building a library of components, we can then provide a lower bound robustness evaluation over the whole search space, rather than single attacks explored as a current practice.

---

> > ### Comment · Reviewer_KocB · 2021-08-17
> > **Good work in progress but results are not strong enough to be convincing**
> >
> > >Can the network transformation handle redundant inputs or outputs?
> > Yes, this can be easily handled by adapting the search space.
> >
> > Can authors please elaborate on this adaptation and why it is easy?
> >
> > > Could you provide ablation studies performed on models (B10-B14)? Why is there not a significant improvement for these models?
> > We are happy to include additional discussion in the revised version of our paper. In short, these models aim to improve the adversarial training procedure rather than developing a new defence. For example, they employ additional unlabelled data for training, extensive hyperparameter tuning, instance weighting or loss regularization. As a result, existing adversarial attacks already provide close to optimal performance.
> >
> > In the opinion of the reviewer, it is important to consider improvement in adversarial training methods as well, since they represent state-of-the-art.
> >
> > > Would the defense algorithms not be able to exploit the knowledge of this grammar?
> > Yes, this is indeed the case and will be the case for any empirical attack that is available to the defence at design time (even if it comes from a very large grammar).
> >
> > Wouldn't it be challenging to use this technique for evaluation when defense methods can design against this? Large size is an attempt to security (in this case, security for attack) through obscurity, and it is not a good idea to have evaluation methods rely on this.
> >
> > The paper appears to be a good work in progress, but lacks sufficiently strong experimental results to convince this reviewer to raise his/her score.

---

> > > ### Author Response · Authors · 2021-08-20
> > > **Clarifications**
> > >
> > > > Can authors please elaborate on this adaptation and why it is easy?
> > >
> > > From the analysis perspective, the model is represented as a computational graph and we can define network transformations directly on the graph (this is similar to how for example Grappler, the tensorflow runtime graph optimization works). In this particular case:
> > >
> > > - To discover redundant outputs, one could replace them with constant value and check whether whether the downstream model results change (i.e., logits or probabilities).
> > > - To discover redundant inputs, one can similarly replace them with constant value.
> > > - Another transformation can simply drop some inputs (or introduce zero inputs) in case the dimensions do not match.
> > >
> > > Even if the output changes, one can still use the transformation to obtain a proxy model to attack. This is possible because while we use the proxy model to find the adversarial perturbation x’, we always evaluate x’ on the original model.
> > >
> > > > it is important to consider improvement in adversarial training methods as well, since they represent state-of-the-art.
> > >
> > > This is exactly what we did by discovering attacks with 2-3x better runtime. The fact that for these methods we did not find significantly stronger attacks is not a limitation of our approach — these models are easier to attack (the robustness comes from training the weights in a better way, not by tweaking the architecture and including explicit defense mechanisms) and that’s why A^3 leads to bigger improvements for more complex defenses.
> > >
> > > As a comparison, we performed an ablation study that compared:
> > > - empirical robustness (considered in our work, which gives lower bound on the non-robust samples) with
> > > - certified robustness that computes robustness with provable guarantees (in our case, we used probabilistic guarantees from [1])
> > >
> > > When considering cifar-10 with L2 norm and eps=0.12, and the model [2] from robust bench, the results are:
> > > - Empirical robustness: 86.8%
> > > - Certified robustness: 86%
> > >
> > > As can be seen, the gap is only 0.8% which also limits the improvement that A^3 can find for such a model.
> > > (The above numbers are illustrative of the robustness, but not necessarily directly comparable due to how the certified robustness is computed)
> > >
> > >
> > > [1] Certified Adversarial Robustness via Randomized Smoothing. ICML’19
> > >
> > > [2]  Improving Adversarial Robustness Using Proxy Distributions. arXiv’21
> > >
> > > > Wouldn't it be challenging to use this technique for evaluation when defense methods can design against this?
> > >
> > > No, it wouldn’t be challenging because this is exactly what the tools was designed for, that is, to:
> > > - Automate tedious and error prone work that is currently done manually when evaluating new defenses,
> > >
> > > Which is exactly what is required when developing a new defense.
> > >
> > > As mentioned in our paper, we are not aiming to replace the domain expert who would still need creativity to design new attacks, but rather provide support for them (in-line with other AutoML and NAS techniques).
> > >
> > > >  Large size is an attempt to security (in this case, security for attack) through obscurity, and it is not a good idea to have evaluation methods rely on this.
> > >
> > > It is important to note that our goal was _not_ to:
> > > - Attempt at security through obscurity, but rather at a systematic way of taking advantage of known approaches. Nor the goal was to,
> > > - Develop a perfect attack that can not be circumvented — this is addressed by concurrent line of work on abstract interpretation and using convex relaxations to provide certified guarantees. However, this line of work has its own limitations, such as scalability to only small networks and errors caused by over-approximations, both of which significantly limit practical use of these methods.

---

### Review · Ethics_Reviewer_DiFU · 2021-08-10

**Recommendation:**

Yes. As mentioned above, the authors are on the right track and can easily address the issues within the paper.

The authors could consider the following recommendations:

1. Add a new impact section between `6. Related Work` and `7. Conclusion`.
2. Port what currently exists in the response to checklist question `1(c)` as a sort of outline for structuring the potential negative societal impact.
3. Expand on the discussion slightly: what are some interesting hypothetical scenarios the authors could foresee where a nefarious actor used their methods to eventually disturb and/or break models in non-trivial ways? To my mind, clear hypothetical scenarios touch on safety/security as well as human rights concerns (for targeted harassment or abuse).
4. Offer potential recommendations and a discussion on risk mitigation opportunities. As mentioned above, I think this could be a very interesting few sentences. One easy step would be an interdisciplinary discussion to better understand what safety procedures or mechanisms have and have not worked in the past in the fields of information/cybersecurity.



**Ethical Issues:**

Yes

**Ethics Review:**

One reviewer flagged this submission for ethics review, and I largely agree with their comments.

According to the NeurIPS ethical guidelines, submissions are to include a discussion about the potential negative societal impacts of the proposed research artifact or application.

In response to 1(c) the authors note that:

>In this paper, an approach to improve the evaluation of adversarial defenses by automatically finding adaptive adversarial attacks is proposed and evaluated. In general, such tool can be used both in a beneficial way by the researches developing adversarial defenses, as well as, in a malicious way by an attacker trying to break existing models. In both cases, the approach is designed to improve empirical model evaluation, rather than providing verified model robusness [sic], and thus is not indended [sic] to provide formal robustness guarantees for safety critical applications.

I agree with the authors' essential assessment: this is a dual-use tool that could easily be used to nefarious, unethical ends to "break existing models," as well as in ways that can prepare adversarial defenses. I believe that the authors should include this discussion within the actual paper itself — especially given the novel problems that such a tool could pose for the broader research community, industry, and governments going forward. I think this section would make a lot of sense between `Related Work` and `Conclusion`. Based on what they've written in the checklist, I believe the authors are on the right track and could easily expand the discussion of potential positive and negative use cases. Such unethical use could either raise safety or security concerns to some extent, and could also potentially be used by an adversary in ways that raise human rights concerns.

According to the NeurIPS ethical guidelines, whenever potential negative social impacts are identified within a paper, "submissions should also include a discussion about how these risks can be mitigated." Based on my reading, this paper does not include discussion of potential mitigation approaches (nor does the checklist statement in `1(c)`). This is easily fixable, however.

As one of the reviewers pointed out, I believe this paper actually offers the authors a great opportunity to introduce a lively (though likely brief) discussion potential mitigation approaches to the stated risks. Other fields (information security, cybersecurity, etc.) have spent years (sometimes making many mistakes and not operating as intended) developing processes for responsible disclosure, bug bounty programs, and more. These processes don't offer a perfect analogy, but I believe they nevertheless offer a good platform to start from. One recommendation this paper could make is dedicated engagement with existing fields who persistently deal with dual-use technology to learn or mechanisms, processes, and policies that could lead to responsible release of the methods contemplated here.

---

> ### Comment · Area_Chair_ifSc · 2021-08-18
> **I disagree with the ethical concerns raised.**
>
> I disagree with the above ethics review. This paper does not do anything that the ~1000 papers on adversarial machine learning, or the tens of thousands of papers in the computer security literature, have done before.
>
> If the reviewer were to take a look at any paper from a top computer security venue (USENIX Security, IEEE S&P, Crypto, CCS or NDSS) they would find that papers that it is commonly understood that papers causing significantly more real-world harm do not address this fact in the text of the paper (cf. Spectre or Meltdown)---because it is well understood that attacks on potential ideas are a necessary component of research. To repeat the boilerplate "attacks are a necessary component of developing strong defenses, and by showing this attack we can help improve security in the future" in every paper would be a waste of everyones' time, and insult the readers' intelligence.
>
> Most importantly, **this paper does not attack any real system**. There is no one to whom the vulnerability can be responsibly disclosed. Further, it develops a technique that can be used to evade machine learning classifiers in a *white-box* scenario (the least practical) and the attack is *strictly less strong* than a human performing a manual adaptive attack.
>
> Consider for analogy when new cryptanalytic techniques are developed to more accurately study the security of various ciphers. When these papers introduce new general tools that can be conceivably applied to any system, they do not go into a discussion of ethics precisely because it is well established that **not** disclosing the attack is the ethically harmful decision. It's well know that the Bad Guys will employ equally smart researchers, and they *won't* publish their findings. So we need researchers to write attacks down in order to allow the Good Guys to build high quality defenses.
>
> If the recommendation made in this ethics review were to be made a policy, every paper in the area of computer security would have to make an identical argument that rehashes the same responsible disclosure discussions that have been going on for the last three decades. It is much better if we just understand that any research papers in the area of computer security---especially those that don't actually break any real deployed system---are necessarily going to contain attacks.

---

> > ### Comment · Ethics_Reviewer_DiFU · 2021-08-28
> > **Response to the objection**
> >
> > Thank you for your raising your concerns regarding both of the ethics reviews. I can't speak for the other reviewer, but I remain a bit confused about how you believe the authors should proceed given the underlying ethics guidelines for submissions to NeurIPS.
> >
> > According to the ethics guidelines: "[s]ubmissions to NeurIPS are expected to include a discussion about potential negative societal impacts of the proposed research artifact or application. (For NeurIPS 2021, this corresponds to question 1c of the NeurIPS Paper Checklist). Whenever these are identified, submissions should also include a discussion about how these risks can be mitigated."
> >
> > In response to the other ethics review, the commenter mentioned that "the authors have adequately discuss the potential negative consequences of their work, and should not be required to modify their paper to add any further discussion on top of their existing statement." But this discussion is in the checklist, not the paper itself. (I also believe the other ethics reviewer has stronger objections than I do regarding the underlying ethical concerns.)
> >
> > My review simply mentioned that their stated potential negative impacts should be surfaced in the paper itself and they should slightly expand upon it and include an interesting discussion of potential mitigations. I do not believe that this would be an insult to readers' intelligence.
> >
> > The commenter also noted that "[i]f the recommendation made in this ethics review were to be made a policy, every paper in the area of computer security would have to make an identical argument that rehashes the same responsible disclosure discussions that have been going on for the last three decades."
> >
> > I'm not trying to create policy, nor be pedantic. I'm trying to faithfully apply the guidelines, and it's more than certainly possible that reasonable minds might disagree about how to approach the negative societal impact statement/discussion inside the paper. Per the guidelines, papers "are expected to include a discussion about potential negative societal impacts," and then "include a discussion about how these risks can be mitigated." Either there are _no_ foreseeable potential negative societal impacts, or the paper should include discussion of those potential societal negative impacts, along with approaches to mitigate those risks.

---

> > > ### Comment · Area_Chair_ifSc · 2021-08-28
> > > **Policy**
> > >
> > > I'm happy to agree that if the NeurIPS policy requires that the discussion appear in the main body of the paper, and not in the checklist response, then the authors should move this paragraph above. I do not think this paper enables anything new that wouldn't be possible with already published work, other than more easily evaluating defenses to adversarial examples.
> > >
> > > So the question here, then, is this: Does a paper that builds on area X, and introduces new contribution Y, have to explain the potential harms of X? Or just the potential harms of Y? It seems to me the policy says the latter. If the policy is actually the former then you are right, and the authors would be required to add extra discussion. (This policy would also be, in my opinion, useless. But let's not have that argument here. If it's the policy then it's the policy.)
> > >
> > > *"the other ethics reviewer has stronger objections than I do"*
> > > The other ethics reviewer makes factually inaccurate statements. We can discuss the problems in that review above if you wish.

---

> > > > ### Author Response · Authors · 2021-09-01
> > > > **Response**
> > > >
> > > > Dear ethics reviewers, dear area chair,
> > > >
> > > > Firstly, thank you for the discussion. We feel that many of the points raised in the discussion are general points that are to be clarified for the NeurIPS and ML commnunity as a whole, rather than points related to our work. Having said that, we are naturally happy to incorporate any changes that would be required to adhere to the officially NeurIPS policy.
> > > >
> > > > One question that was not clear is where the discussion should appear -- should it be in the main body, checklist response, after the main text (before bilbliography)? We've reviewed the ethics guidelines available at :
> > > >
> > > > https://neurips.cc/public/EthicsGuidelines
> > > >
> > > > where it says that:
> > > >
> > > > > (For NeurIPS 2021, this corresponds to question 1c of the NeurIPS Paper Checklist)
> > > >
> > > > In our understanding means that the reponse should go into the question 1c in checklist.

---

### Review · Ethics_Reviewer_m2wu · 2021-08-12

**Recommendation:**

As stated above, the issue is not simply with this paper, but all such papers which discover or develop automatic attacks. Within this paper, I think the authors can help address the ethical concerns by:
1) Provide more of an explanation for how actors can easily use their work to develop better and more robust defenses
2) Acknowledge the serious concerns
3) As experts in the domain, perhaps suggest future ways the community can work together to mitigate malicious uses of these types of paper.

Fundamentally I'm not sure that it's possible o completely mitigate the ethical concerns in this type of paper.

**Ethical Issues:**

Yes

**Ethics Review:**

By developing capabilities to automatically identify attacks that can defeat even advanced defense capabilities, this paper is necessarily raising serious ethical concerns. Such technology can be leveraged by bad actors to threaten the safety and security of infrastructure (by enabling ransomware and other such attacks), cause a detrimental effect on people’s livelihood or economic security (by stealing encrypted or protected information), or engage in harmful forms of surveillance (by defeating security defenses on communication platforms).

Most troubling is that the potential deleterious effects (e.g. using an automatic attack framework to automatically make attacks) is a more plausible use case then the pro-social use case (e.g. using an automatic attack framework to help improve defenses). This isn't to say that the latter is impossible, but rather that it's practically and conceptually harder to use this paper for a non-malicious purpose.

It's unfortunate, since the primary ethical critique applies to any such paper, including many that the authors cite. Thus it is not this paper in particular that is trouble, but rather the entire genre of papers that explore this technique.

---

> ### Comment · Area_Chair_ifSc · 2021-08-18
> **I disagree with the ethical concerns raised.**
>
> I disagree with the above ethics review. The paper does not cause any of the three harms the reviewer lists.
>
> I do not see how anything technical introduced in this paper allows for "enabling ransomware" or "stealing encrypted or protected information". This paper introduces a technique to evade machine learning classifiers in a white-box setting. This has nothing to do with ransomware, and even less to do with stealing protected content.
>
> Further, this paper actually *reduces the likelihood of surveillance*. If a surveillance tool were to use machine learning (e.g., to perform ASR to transcribe everything that was being discussed at all times over all calls), an adversarial attack would allow two parties to communicate *without* being surveilled. An attack here does not help Big Brother--it makes their job harder.
>
> As such, the authors have adequately discuss the potential negative consequences of their work, and should not be required to modify their paper to add any further discussion on top of their existing statement that "In general, such tool can be used both in a beneficial way by the researches developing adversarial defenses, as well as, in a malicious way by an attacker trying to break existing models. In both cases, the approach is designed to improve empirical model evaluation, rather than providing verified model robustness, and thus is not intended to provide formal robustness guarantees for safety critical applications."

---

### Decision · Program_Chairs · 2021-09-27

**Decision:**

Accept (Poster)

**Comment:**

This paper introduces a technique to better perform adaptive attacks on adversarial example defenses. Current evaluations are large done by hand, and the paper here introduces a technique to automate some of that process. While the reviewers all like the overall tone of the paper, the reviewers are generally concerned about three aspects of the paper.

First, the paper does not demonstrate that it can automate all possible hand-specified attacks (for example, it has a limited design space that might not identify redundant neurons).

Second, the paper considers only a subset of the attacks used in prior papers. And so saying that this attack can match prior work may not actually be valid.

Third, the attack may not yet be practical to use for defenders.

Taken as a while, while I agree that these are definite limitations to this technique, this is the first paper to try these kinds of automated adaptive attacks and is worth publishing. While the implementation is imperfect at the moment, and is more work to be done on evaluation and improved attack, this can be done with followup work. If it turns out that this idea is not actually better than the human-designed attacks, future work could help identify the cases where this is true or improve the attack to make it more effective. Even though this attack is not likely practically useful for defenders yet, the new direction it opens might help in the future.